# Structure of human steroid 5α-reductase 2 with the anti-androgen drug finasteride

Qingpin Xiao[1,2], Lei Wang [3], Shreyas Supekar [4], Tao Shen [5], Heng Liu[3], Fei Ye[5], Junzhou Huang[5], Hao Fan [4✉], Zhiyi Wei [1✉] & Cheng Zhang [3✉]

Human steroid 5α-reductase 2 (SRD5A2) is an integral membrane enzyme in steroid metabolism and catalyzes the reduction of testosterone to dihydrotestosterone. Mutations in the *SRD5A2* gene have been linked to 5α-reductase deficiency and prostate cancer. Finasteride and dutasteride, as SRD5A2 inhibitors, are widely used antiandrogen drugs for benign prostate hyperplasia. The molecular mechanisms underlying enzyme catalysis and inhibition for SRD5A2 and other eukaryotic integral membrane steroid reductases remain elusive due to a lack of structural information. Here, we report a crystal structure of human SRD5A2 at 2.8 Å, revealing a unique 7-TM structural topology and an intermediate adduct of finasteride and NADPH as NADP-dihydrofinasteride in a largely enclosed binding cavity inside the transmembrane domain. Structural analysis together with computational and mutagenesis studies reveal the molecular mechanisms of the catalyzed reaction and of finasteride inhibition involving residues E57 and Y91. Molecular dynamics simulation results indicate high conformational dynamics of the cytosolic region that regulate NADPH/NADP$^+$ exchange. Mapping disease-causing mutations of SRD5A2 to our structure suggests molecular mechanisms for their pathological effects. Our results offer critical structural insights into the function of integral membrane steroid reductases and may facilitate drug development.

[1] Department of Biology, Southern University of Science and Technology, 518055 Shenzhen, Guangdong, China. [2] Faculty of Health Sciences, University of Macau, 999078 Macau, SAR, China. [3] Department of Pharmacology and Chemical Biology, School of Medicine, University of Pittsburgh, Pittsburgh, PA 15261, USA. [4] Bioinformatics Institute (BII), Agency for Science, Technology and Research (A*STAR), Singapore 138671, Singapore. [5] Tencent AI Lab, 518000 Shenzhen, Guangdong, China. ✉email: fanh@bii.a-star.edu.sg; weizy@sustech.edu.cn; chengzh@pitt.edu

The membrane-embedded 5α-reductase (steroid 5α-reductase; SRD5A) family in humans includes five members, SRD5A1–3 and the much less thoroughly characterized glycoprotein synaptic 2 (GSPN2) and GSPN2 like[1]. They mainly catalyze the irreversible reduction of the $\Delta^{4,5}$ bond in $\Delta^4$-3-ketosteroids using reduced nicotinamide adenine dinucleotide phosphate (NADPH) as the hydride donor cofactor[1,2], although other lipid substrates, such as polyprenols have also been identified as substrates of SRD5A3 (ref. [3]). SRD5As are expressed differently in the human body to play diverse functional roles despite their sequence similarity (Extended Data Fig. 1). SRD5A1 and SRD5A3 have been indicated to function in the metabolism of neurosteroids[4,5] and in protein N-linked glycosylation[3,6], respectively. SRD5A2 is the most intensively investigated SRD5A, and has well-characterized roles in androgen metabolism and androgen-related disorders[1,7]. All three SRD5As are located in the membrane of endoplasmic reticulum (ER) in the cells[8].

SRD5A2 is highly expressed in male reproductive systems[7] to convert testosterone to 5α-dihydrotestosterone (DHT; Fig. 1a), the major steroid hormone for androgen receptor[1]. A large number of mutations identified in the *SRD5A2* gene can result in insufficient levels of DHT, leading to an autosomal recessive disorder named 5α-reductase deficiency, which is associated with underdeveloped and atypical genitalia[9–11]. On the other hand, overproduction of DHT by SRD5A2 is associated with benign prostatic hyperplasia (BPH), androgenic alopecia, and prostate cancer due to excessive androgen receptor signaling[7,12]. 5α-Reductase inhibitors (5ARIs) including finasteride and dutasteride (Fig. 1b), which mainly target SRD5A2, but also act on other SRD5As[13], have been used as a major class of antiandrogenic drugs to treat BPH and androgenic alopecia[1,7,12,14], and are indicated in the treatment of prostate cancer[15]. In particular, finasteride is among the top 100 most prescribed drugs in the United States and is associated with an irreversible action on SRD5A2 (refs. [16,17]). Interestingly, androgen receptor signaling can lead to the expression of transmembrane serine protease 2, which is required for the entry of SARS-CoV-2 and other coronaviruses into host cells[18,19]. Therefore, androgen signaling has recently been linked to COVID-19 disease severity, explaining why males are more prone to severe COVID-19 symptoms[20]. The 5ARI drugs that can significantly reduce androgen signaling have thus been suggested to hold potential for repurposing to treat COVID-19 (refs. [20,21]).

SRD5As belong to a large group of eukaryotic membrane-embedded steroid reductases, which also include sterol reductases, such as 7-dehydrocholesterol reductase (DHCR7) that catalyzes the last step in cholesterol biosynthesis in humans[22]. Although these steroid/sterol reductases share very little sequence similarity, they all use NADPH as the cofactor to reduce specific carbon–carbon double bonds in their steroid substrates. To date, only one crystal structure of a bacterial membrane-embedded sterol reductase MaSR1 without any steroid substrate has been reported for this group of reductases[23]. To further understand the molecular mechanisms underlying the function of eukaryotic steroid reductases and, in particular, the catalytic mechanism of SRD5As and the action of 5ARI drugs, we solved a crystal structure of human SRD5A2 in the presence of NADPH and finasteride. The structure revealed a topology of seven transmembrane α-helices (7-TMs), rather than the 10-TM topology of MaSR1, and an NADP–dihydrofinasteride (NADP–DHF) intermediate adduct. This structure together with computational studies provided detailed molecular insights into the catalytic mechanism of SRD5A2, the irreversible action of finasteride on SRD5A2, and the molecular mechanisms underlying the pathological effects of disease-associated mutations.

## Results

**Structure determination and overall structure of human SRD5A2.** We expressed human SRD5A2 in insect Sf9 cells. Initially, we tried to purify it without any ligand and found that most of the purified protein aggregated. This result was consistent with previous studies showing a rapid loss of enzyme activity for purified SRD5A2 (refs. [24–26]). We speculated that the ligand-dependent stabilization of SRD5A2 may be important for protein purification and crystallization, similar to G protein-coupled receptors (GPCRs)[27]. We then purified the enzyme in the presence of finasteride (see "Methods"), and the results showed a single and monomeric peak in size-exclusion chromatography (SEC; Extended Data Fig. 2a). The purified SRD5A2 with finasteride was then crystallized in lipidic mesophase[28,29] with a space group of *P*622 (Extended Data Fig. 2). The structure was determined to 2.8 Å resolution by molecular replacement[30] based on a structural model from de novo prediction, since we failed to use anomalous diffraction data to solve the structure (see "Methods" and Extended Data Fig. 3). A similar evolutionary coupling analysis approach has been used to solve the crystal structure of the bacterial peptidoglycan polymerase RodA[31]. Clear electron density allowed modeling of all 254 residues of SRD5A2 except for the first four residues and residues S39–A43 in a flexible loop region (Fig. 1c, d and Table 1). A dualsteric ligand that occupies two different binding sites[32] was modeled as an adduct of

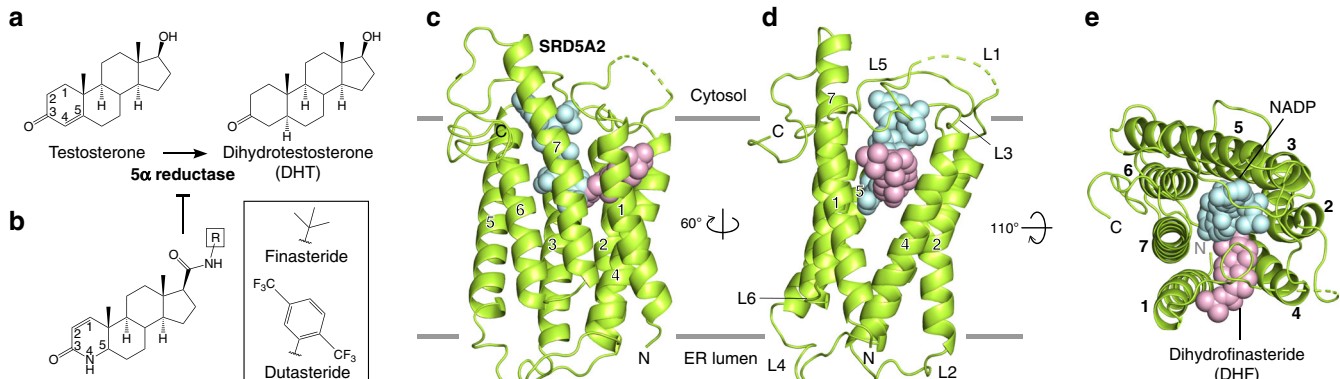

**Fig. 1 Overall structure of human SRD5A2. a** 5α-reduction reaction of the $\Delta^{4,5}$ double bond of testosterone catalyzed by SRD5A2 to generate dihydrotestosterone (DHT). **b** SRD5A2 inhibition by finasteride and dutasteride. The two inhibitors share the same ring structure with different R-groups connected to the amide side chains. **c–e** Three views of the SRD5A2 structure. The NADP–DHF adduct was shown as spheres. L1–6 represent six loops connecting 7-TMs. The NADP and DHF moieties were colored in light cyan and light pink, respectively.

**Table 1 X-ray data collection and refinement statistics (molecular replacement).**

|  | SRD5A2 with NADP–DHF[a] |
|---|---|
| **Data collection** |  |
| Space group | P622 |
| Cell dimensions |  |
| $a, b, c$ (Å) | 107.4, 107.4, 103.4 |
| $\alpha, \beta, \gamma$ (°) | 90, 90, 120 |
| Resolution (Å) | 50–2.8 (2.85–2.8)[b] |
| $R_{merge}$ | 0.248 (1.274) |
| $I/\sigma I$ | 13.0 (1.05) |
| Completeness (%) | 99.7 (100.0) |
| Redundancy | 9.6 (6.8) |
| **Refinement** |  |
| Resolution (Å) | 50–2.8 (3.19–2.8) |
| No. reflections | 9155 (2857) |
| $R_{work}/R_{free}$ | 0.239 (0.297)/0.265 (0.336) |
| No. atoms | 2036 |
| Protein | 1926 |
| Ligand/ion | 110 |
| Water | 0 |
| B-factors |  |
| Protein | 76.4 |
| Ligand/ion | 66.5 |
| R.m.s. deviations |  |
| Bond lengths (Å) | 0.003 |
| Bond angles (°) | 0.77 |

[a]Diffraction data of five crystals were merged.
[b]Values in parentheses are for highest-resolution shell.

finasteride and NADPH in the structure (Fig. 1e), which will be discussed later.

Unlike the bacterial sterol reductase MaSR1 (ref. [23]), the structure of SRD5A2 contains 7-TMs (TM1–7) connected by six loops (L1–6; Fig. 1c, d and Extended Data Fig. 4a). We assigned the carboxyl terminal (C-terminal) loop (C-loop) to face the cytosol and the amino terminal loop (N-loop) to face ER lumen according to the enriched positively charged residues at the C-terminal side[33] (Extended Data Fig. 4b). In addition, the N-terminal residue C5[N] (superscripts indicate the locations of the residues hereafter) forms a disulfide bond with C133[L4] in loop 4 (L4) (Extended Data Fig. 4a), suggesting an ER luminal location of the N-terminal side because of the reducing environment of the cytosol. 7-TM topology is more commonly associated with GPCRs[34], although the arrangement of TMs in SRD5A2 is distinct from that of GPCRs (Extended Data Fig. 4c).

**Intermediate adduct formed between finasteride and NADPH.** The structure of SRD5A2 revealed a large cavity inside the 7-TM domain at the cytosolic side formed by all 7-TMs and L1, L3, and L5 (Fig. 1c, d). The cavity is completely occluded from the cytosol with only one opening on the side of the 7-TM domain between TM1 and TM4. Clear electron density in the cavity revealed features of NADPH and finasteride, which allowed unambiguous modeling of both ligands (Fig. 2a). Strikingly, after ligand fitting, the distance between the nicotinamide C-4 atom of NADPH and the C-2 atom of finasteride is ~1.5 Å, suggesting the formation of a covalent bond (Fig. 2a, b).

Strongly supporting our structural findings, a previous enzymological and mass spectrometric study on the mechanism of SRD5A2 inhibition by finasteride identified an intermediate adduct as NADP–DHF[17]. The same study indicated that SRD5A2 could catalyze the hydride transfer from NADPH to finasteride, leading to the formation of a covalent bond between the nicotinamide C-4 atom of NADPH and the C-2 atom of

finasteride[17], which is highly consistent with our structural observation. It is likely that SRD5A2 used endogenous NADPH during recombinant protein expression to catalyze the reaction with finasteride supplemented in protein buffers to generate the NADP–DHF intermediate, which was stable enough to be captured in our SRD5A2 crystals. In fact, NADP–DHF has been suggested to be one of the most potent noncovalent enzyme inhibitors in general with reported binding constant $K_i \leq 3 \times 10^{-13}$ M and off-rate $k_{off} = 2.74 \times 10^{-7}$ s$^{-1}$ (ref. [17]), which means that the half-life of the SRD5A2 and NADP–DHF complex is ~31 days, explaining the irreversible action of finasteride on SRD5A2. Therefore, we modeled NADP–DHF as the real ligand in our structure.

**Binding pockets for NADP–DHF.** The substrate-binding cavity shows two relatively separate tunnel-like pockets for NADP and DHF (Fig. 2c). NADP adopts an extended anti-conformation to insert into the binding pocket with a positively charged environment inside the 7-TM bundle (Fig. 2c). Surprisingly, the binding pocket for NADP is enclosed on the cytosolic side by the cytosolic loops, completely shielding NADP from the cytosol (Fig. 1e). The nicotinamide-ribose moiety is buried inside 7-TM and stabilized by extensive polar and hydrophobic interactions with residues mainly from TMs 2 and 4–7, while the diphosphate moiety of NADP mainly forms polar interactions with L1 and TM6–7 (Extended Data Fig. 5a–d). The adenine-ribose phosphate moiety of NADP forms hydrogen bonds and salt bridges with residues mostly from three cytosolic loops (Extended Data Fig. 5a–d). Supporting this cofactor-binding mode, mutations of residues that interact with NADP have been shown by previous enzymological studies to either decrease or abolish the catalytic activity of SRD5A2 (Extended Data Fig. 5e)[35–37].

In contrast to the highly polar environment of NADP, the binding pocket of DHF is largely hydrophobic, where the core ring structure of DHF interacts with hydrophobic residues from TMs 1, 2, 4, and 7 (Fig. 2d and Extended Data Fig. 5d). A previous study showed that substitution of one aromatic residue, F118[TM4], in the DHF-binding pocket to a leucine could dramatically decrease the SRD5A2 activity by disrupting the binding of testosterone[38], suggesting an important role of this residue in the binding of steroid substrates. The polar groups located at each end of DHF engage in additional polar interactions with residues E57[TM2] and R114[TM4] of SRD5A2 (Fig. 2d). Structures of several GPCRs with sterol ligands, such as Smoothened[39] and the bile acid receptor GPBAR[40] have been reported. The sterol ligands in these GPCRs insert into the transmembrane domain perpendicularly to the membrane, in contrast to the binding pose of DHF, which is almost parallel to the membrane. No obvious conserved features in the recognition of sterol ligands are observed for these GPCRs and SRD5A2, suggesting highly diverse mechanisms of sterol ligand recognition by membrane proteins.

Sequence alignment analysis of SRD5As across different species, including the plant homolog DET2 involved in the synthesis of phytohormones[41], indicated that while the residues involved in the binding of NADP are highly conserved, the residues in the DHF-binding pocket are much less conserved (Extended Data Fig. 1). This suggests that although all the SRD5A family members use NADPH as the cofactor to reduce their substrates, they have evolved specific structural features to recognize different steroid or lipid substrates[1,7]. We further mapped the residue conservation to the SRD5A2 structure, which showed that the region around the tert-butylacetamide tail group of DHF is the least conserved part of the ligand-binding pocket (Extended Data Fig. 5f), explaining the selectivity of finasteride

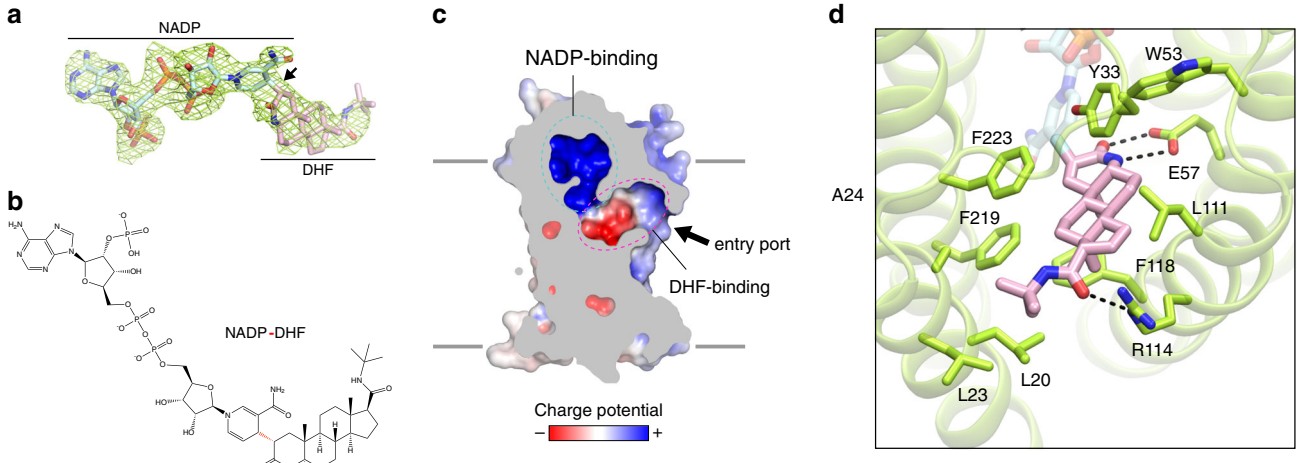

**Fig. 2 Formation and binding environment of NADP–DHF adduct. a** $F_o$–$F_c$ electron omit map of the NADP–DHF adduct contoured at 3σ. **b** Chemical structure of the NADP–DHF adduct. The covalent bond connecting the nicotinamide C-4 atom of NADPH and the C-2 atom of finasteride is highlighted in red. **c** Enclosed binding cavity for NADP–DHF with charge potentials. The potential entry port for the steroid substrates is indicated by an arrow. Membrane boundaries are shown as gray lines. **d** Molecular details of the DHF-binding pocket. Hydrogen bonds (2.2–3.2 Å) are indicated by dashed lines.

for human SRD5A2 over SRD5A1 (refs. [1,14]). Specifically, R114 of SRD5A2 forms a hydrogen bond with the tert-butylacetamide tail group of DHF (Fig. 2d). This residue is replaced by a methionine residue in SRD5A1 (Extended Data Fig. 1), potentially disrupting the hydrogen bond with finasteride.

**Potential mechanisms of SRD5A2 catalysis and inhibition.** It has been proposed that two unknown residues in SRD5A2 point toward the C-3 carbonyl group in finasteride to facilitate the hydride transfer from NADPH to finasteride to form an enolized intermediate, which is followed by the formation of a covalent bond between finasteride and NADP[17]. In our structure, both the C-3 carbonyl and the N-4 amine groups of DHF form hydrogen bonds with E57[TM2] (Fig. 3a). We propose that through hydrogen-bonding interactions with E57[TM2], finasteride is positioned in a way that the 4-pro-(R)-hydride of NADPH is in the proximity of the C-2 atom of finasteride to allow hydride transfer to the $\Delta^{1,2}$ bond in finasteride. Due to the presence of the N-4 amine group, the enolization of finasteride as the result of hydride transfer involves the C-3 carbonyl and the C-2 group, leading to the covalent bond formation between finasteride and NADP (Fig. 3b). Dutasteride contains the same core ring structure as finasteride (Fig. 1b), and therefore likely forms a similar adduct with NADPH. Indeed, dutasteride has been suggested to share the same irreversible inhibition mechanism as finasteride[42].

Interestingly, finasteride also inhibits the activity of steroid 5β-reductase, a soluble steroid reductase that belongs to the NADPH-dependent aldo-keto reductase (AKR) superfamily[43]. Steroid 5β-reductase, also named AKR1D1, can reduce the $\Delta^{4,5}$ bond in testosterone as SRD5A2 does, but generates a stereochemically different product, 5β-DHT[2]. A crystal structure of AKR1D1 with finasteride[44] showed that the relative position of finasteride to NADP+ in AKR1D1 is opposite to that of DHF to NADP in SRD5A2 (Extended Data Fig. 5g). As a result, the 4-pro-(R)-hydride of NADPH is adjacent to the N-4 group instead of the C-2 group of finasteride (Extended Data Fig. 5g), thereby preventing hydride transfer[44]. Hence, the $\Delta^{1,2}$ bond of finasteride cannot be reduced by AKR1D1, which accounts for the competitive and reversible action of finasteride on AKR1D1 (ref. [44]).

To investigate the binding pose of testosterone and the catalytic mechanism of SRD5A2, we docked NADPH and testosterone to our structure in silico. In the docked structure, while the NADPH

molecule could be well aligned to the NADP moiety of NADP–DHF in the crystal structure (RMSD = 0.58 Å for heavy atoms), testosterone is positioned more deeply into the cavity compared to DHF with its ring structure stacked parallel to the nicotinamide ring of NADPH (Fig. 3c). This is likely due to the absence of the hydrogen bond between E57[TM2] and the N-4 group of DHF. As a result, the C-3 carbonyl group of testosterone forms hydrogen bonds with both E57[TM2] and Y91[TM3], and the 4-pro-(R)-hydride of NADPH is close to the $\Delta^{4,5}$ bond of testosterone (~2.5 Å; Fig. 3c). Previous structural studies on the soluble steroid reductase AKR1D1 have shown that the steroid C-3 carbonyl formed hydrogen bonds with residues E120 and Y58. In ARK1D1, Y58 has been suggested to function as the general acid–base catalytic group to polarize the steroid C-3 carbonyl group together with E120 to facilitate hydride transfer and substrate enolization[45–47]. We propose that SRD5A2 employs a similar catalytic mechanism, in which residues E57[TM2] and Y91[TM3] polarize the C-3 carbonyl of testosterone by hydrogen bonding to facilitate hydride transfer from NADPH to the C-5 atom of testosterone, leading to the formation of an enolized intermediate followed by reduction of the $\Delta^{4,5}$ bond in testosterone (Fig. 3d). Consistently, the optimum pH for the SRD5A2 catalytic activity is acidic (pH~5), which favors the protonation of E57[TM2] to form a hydrogen bond with the C-3 carbonyl of testosterone[35]. Our mutagenesis studies showed that the replacement of E57[TM2] with a less acidic glutamine residue compromised enzyme activity (Fig. 3e and Extended Data Fig. 5h). Likewise, the Y91F mutation, which eliminates the potential hydrogen bonding of Y91[TM3] to testosterone, essentially abolished the conversion of testosterone to DHT by SRD5A2 in our experiments (Fig. 3e and Extended Data Fig. 5h), supporting our proposed catalytic mechanism (Fig. 3d).

Despite sharing a potentially similar catalytic mechanism, the relative orientations of the steroid substrates to NADPH are distinct in SRD5A2 and AKR1D1. In our crystal structure, the nicotinamide ring of NADP is oriented toward the α-face of DHF, and residues E57[TM2] and Y91[TM3] are located on the other side of the core ring structure (Fig. 3a, c). In contrast, in AKR1D1, the nicotinamide ring of NADP+ is oriented toward the β-face of steroid substrates (Fig. 3f)[44–46]. This steroid-binding mode in SRD5A2 suggests that the hydride is transferred from NADPH to the C-5 atom of testosterone at the α-face, leading to the 5α-stereochemistry of DHT generated by SRD5As.

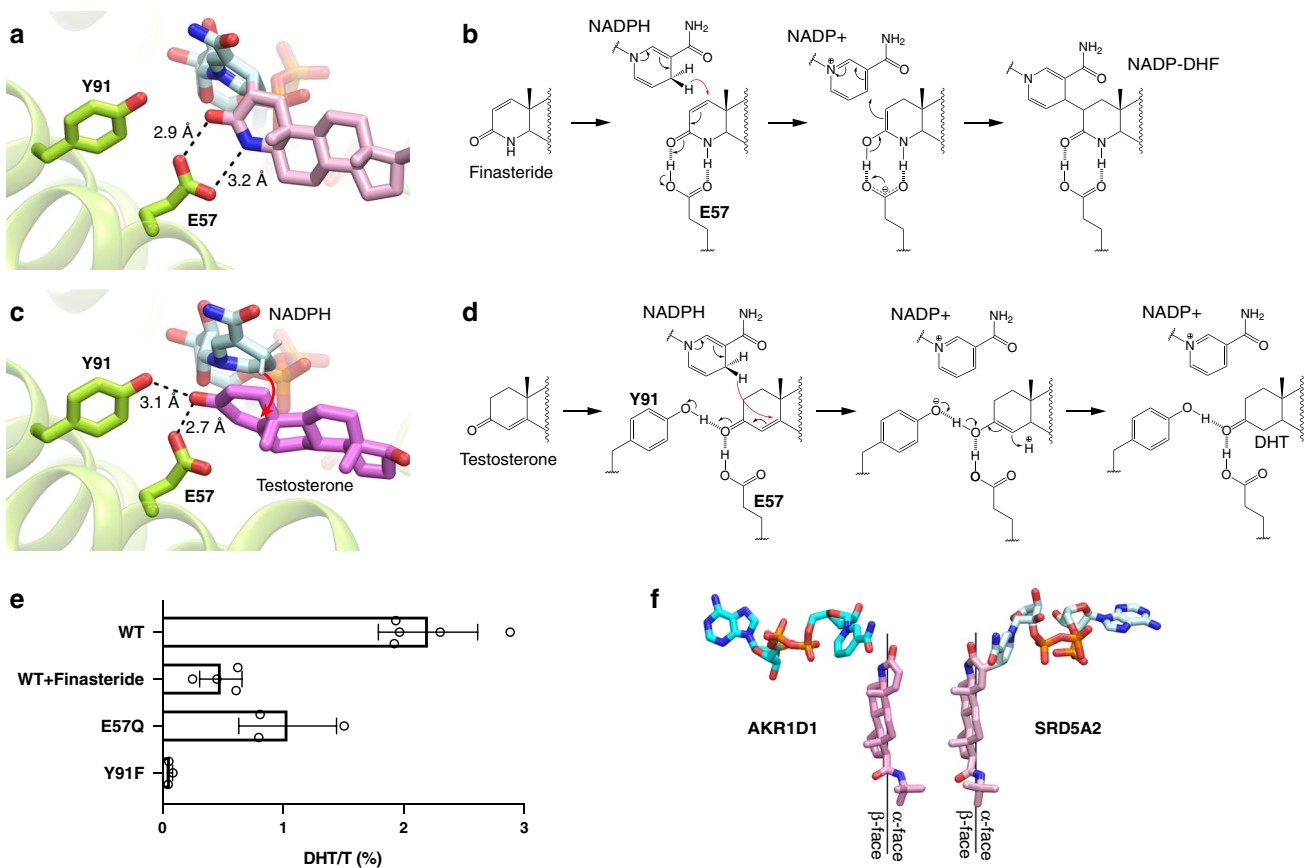

**Fig. 3 Mechanisms of SRD5A2 catalysis and inhibition. a** Binding pose of DHF. Y91[TM3] is not directly hydrogen bonded to DHF. **b** Potential mechanism of finasteride inhibition and the covalent adduct formation between NADPH and finasteride. E57[TM2] facilitates the hydride transfer to the $\Delta^{1,2}$ bond of finasteride, leading to the formation of a covalent bond. **c** Binding pose of testosterone based on our docking results. E57[TM2] and Y91[TM3] each forms a hydrogen bond with the substrate. **d** Potential mechanism the 5α-reduction of testosterone. E57[TM2] and Y91[TM3] facilitate hydride transfer to the $\Delta^{4,5}$ bond of testosterone, leading to the formation of DHT. Hydrogen bonds are shown as dashed lines and the hydride transfer is shown as red curved arrows. **e** Catalysis of testosterone (T) to dihydrotestosterone (DHT) by wild-type SRD5A2 (WT), WT with 500 μM finasteride, and two SRD5A2 mutants E57Q and Y91F. The ratios of DHT to T (DHT/T) were determined by mass spectrometry. All data are presented as the mean ± SEM of three (for E57Q and Y91F), four (for WT + finasteride), and five (for WT) independent experiments. Source data are provided as a Source data file. **f** Distinct orientations of finasteride relative to NADPH in AKR1D1 and SRD5A2. The finasteride and NADPH conformations in the AKR1D1 structure (PDB ID: 3G1R) are shown by aligning the core ring of finasteride to that of DHF in the SRD5A2 structure.

**Structural dynamics of SRD5A2 for catalysis**. The binding pocket of NADP–DHF opens only on the side of 7-TM, allowing steroid substrates access SRD5A2 from the lipid bilayer (Fig. 2c). The cytosolic loops L1, L3, and L5 pack against each other to fully enclose the binding pocket for NADP (Fig. 1d, e), in contrast to the highly exposed NADP$^+$/NADPH-binding pockets in soluble AKRs and MaSR1 (refs. [23,48]). All cytosolic loops are involved in interactions with the adenine-ribose moiety of NADP. Such a conformation is compatible with the very tight binding of NADP–DHF to the enzyme[13], while imposing a physical barrier for NADPH/ NADP$^+$ exchange during the reaction. It is unlikely that the nucleotides either enter or exit from the enzyme through the opening between TM1 and TM4 on the side of 7-TM considering the surrounding lipid environment and the highly polar nature of NADP$^+$/NADPH (Fig. 2c). To lift the barrier, the cytosolic loops in SRD5A2 may undergo conformational changes during the reaction so that the cytosolic region can open up to expose the nucleotide-binding pocket to the cytosol before and after one reaction to allow NADP$^+$/NADPH exchange, and thus efficient reaction turnover.

To further investigate the conformational dynamics of SRD5A2, we performed molecular dynamics (MD) simulations of SRD5A2 with NADP$^+$ (nap) and without NADP–DHF (apo)

on microsecond timescales. The unresolved residues 1–4 at the N-terminus and 39–43 in L1 were incorporated into the SRD5A2 models in simulations (see "Methods"). Principal component analysis (PCA) of the MD trajectories clearly indicated high structural dynamics of the cytosolic loop L1 and to a lesser extent L5, which indeed resulted in the opening of the nucleotide-binding pocket during both nap and apo simulations (Fig. 4a and Extended Data Fig. 6a). In addition, L1 exhibited more pronounced conformational fluctuations in the apo simulations than in the nap simulations (Extended Data Fig. 6a), presumably due to the stabilization of the cytosolic loops by NADP$^+$ in the nap simulations (Fig. 2d and Extended Data Fig. 5a). Since L1 is involved in the binding pockets for both NADP and finasteride in the crystal structure, our simulation results suggest that L1 may function as a "gate" domain to control NADPH/NADP$^+$ exchange and the binding of steroid substrates (Fig. 4b). Consistently, we observed higher B-factors in general for residues in L1 than for residues in the TMs and in other cytosolic loops in the crystal structure, supporting the highly flexible feature of L1 (Extended Data Fig. 6b).

**Disease-related mutations of SRD5A2**. To date, over 100 genetic mutations in the *SRD5A2* gene have been identified according to

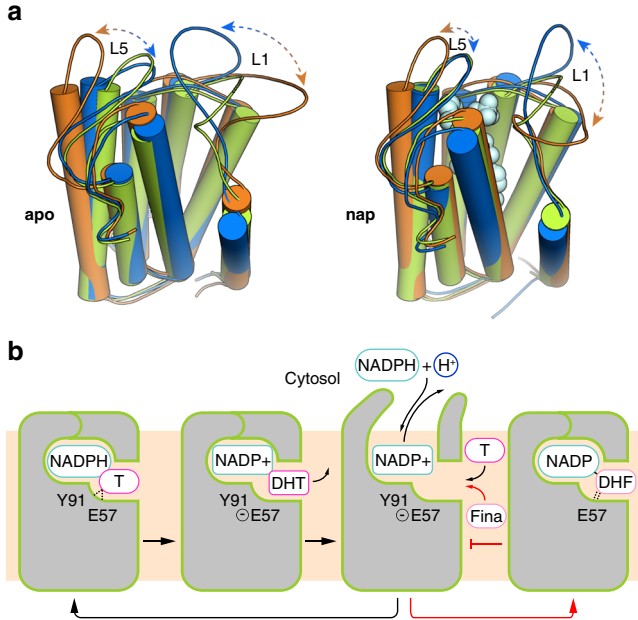

**Fig. 4 Dynamics of SRD5A2 during reaction. a** Dominant conformational motions in the SRD5A2 MD simulations using principal component analysis (PCA). To visualize the motions, two representative conformations from the principal components for the apo and nap states are indicated in orange and blue, respectively. The SRD5A2 crystal structure colored in green is superimposed as a reference. PCA clearly indicates large-scale motions in L1 and L5, suggesting opening up of the nucleotide-binding cavity. **b** Model of the dynamics of SRD5A2 in one cycle of reaction. The reaction can be inhibited by finasteride by forming a stable adduct with the NADPH cofactor to stabilize the closed conformation of SRD5A2. Testosterone and finasteride are indicated as "T" and "Fina", respectively. Testosterone catalysis and the finasteride inhibition are shown with black and red arrows, respectively.

the Human Gene Mutation Database (www.hgmd.cf.ac.uk) to cause the rare autosomal recessive disorder 5α-reductase deficiency[9,10]. Most mutations either abolish or reduce the activity of SRD5A2, leading to significantly reduced levels of DHT in vivo[10]. A majority of 5α-reductase deficiency-causing mutations are missense mutations that are located throughout the whole protein (Fig. 5a). By mapping reported missense mutations of the SRD5A2 protein in our structure, we found that most of the mutation sites are at the ligand-binding cavity (Fig. 5b), suggesting that many of those mutations compromise the activity of SRD5A2 by impairing cofactor/substrate binding or the catalytic process. Examples include two of the founder mutations for the 5α-reductase deficiency patients[10], R227Q and R171S, which diminish SRD5A2 activity, likely by disrupting the hydrogen bonds with NADP (Extended Data Fig. 5c–e). In addition, supporting the critical roles of E57[TM2] and Y91[TM3] in our proposed catalytic mechanism, the mutations E57Q and Y91D have been shown to significantly reduce enzyme activity[35,49]. In addition, some mutations that are not in the ligand-binding cavity presumably impair protein folding/stability. For example, the C133G mutation eliminates the C5–C133 disulfide bridge that links TM1 and TM4, which may be important for the overall folding of SRD5A2 (Fig. 5c). As the most frequently reported mutation site, R246[C] in the C-loop forms multiple hydrogen bonds with residues in L5 (Fig. 5d), potentially stabilizing L5 in position for cofactor binding. Consistently, the R246W mutation showed decreased NADPH-binding affinity in previous studies[9,35].

## Discussion

In contrast to the well-understood mechanisms for the function of soluble steroid reductases, for which numerous structures have been reported[48], the molecular mechanisms governing the function of eukaryotic membrane-embedded steroid reductases have remained enigmatic due to limited structural information. To our knowledge, the reported structure of human SRD5A2 with the dualsteric ligand NADP–DHF represents the first structure of eukaryotic membrane-embedded steroid reductase. Together with computational studies, our structure unveils the binding cavity inside the 7-TM bundle for NADPH and steroid substrates with flexible cytosolic loops. Structural analysis and mutagenesis studies suggest the molecular mechanisms for enzyme catalysis and inhibition involving newly identified residues E57[TM2] and Y91[TM3]. Because of the chemical differences in the steroid core ring, testosterone and finasteride adopt different binding poses relative to NADPH, so that the $\Delta^{4,5}$ bond in testosterone and the $\Delta^{1,2}$ bond in finasteride can be reduced by SRD5A2 to generate distinct end products. Our results explain the 5α-reduction reactions catalyzed by SRD5As, contrasting with the 5β-reduction reactions catalyzed by soluble steroid reductases. The mechanism-based irreversible action of finasteride and dutasteride has led to their successful use as antiandrogen drugs, which may be repurposed for treating COVID-19 patients with excessive androgen receptor signaling[20,21]. Mapping disease-related mutations of SRD5A2 to our structure also provides feasible molecular mechanisms for the effects of those mutations in 5α-reductase deficiency.

The SRD5A2 structure together with simulation studies reveal unexpected structural features for the binding of NADP+/NADPH and steroid substrates. The cytoplasmic loops L1, L3, and L5 enclose the binding pocket of NADPH inside the 7-TM bundle of SRD5A2 to position it close to testosterone for catalysis. To our knowledge, such a structural feature has not been observed in other enzymes using pyridine nucleotides, including NADH and NADPH as cofactors. Our MD simulation results suggest that the cytoplasmic loop L1 undergoes dramatic conformational changes during the reaction to allow NADP+/NADPH exchange (Fig. 4a). The possible large energy barrier for the cytosolic region to overcome to open up the NADPH-binding pocket may hinder NADPH binding. As a result, the reported dissociation constant of NADPH for SRD5A2 (~3–10 μM)[1] is higher than that of NADPH for the soluble steroid reductase AKR1D1, with a highly exposed NADPH-binding pocket (~0.5 μM)[46]. The steroid substrates likely access the ligand-binding pocket of SRD5A2 from the lipid bilayer through the opening between TM1 and TM4 (Figs. 2c, 6a), which is analogous to the lateral ligand entrance mechanism for several GPCRs with lipid ligands[50,51].

Despite sharing very little sequence similarity and having different structural topology, human SRD5A2 and bacterial MaSR1 can be aligned on their core structure of six transmembrane helices (TM2–7 in SRD5A2) that participates in the binding of NADPH and substrates (Fig. 6a). There is also an opening between TM7 and TM10 in MaSR1 (Fig. 6a), which may serve as a potential entry port for its substrates. Such conserved structural features imply that these two enzymes and likely other membrane-embedded steroid reductases take NADPH from the cytosol through cytoplasmic loops, and steroid substrates from the lipid bilayer into ligand-binding cavities inside the cell membrane for catalysis. In contrast to the buried NADP in SRD5A2, the partially modeled NADPH in the structure of MaSR1 occupies a different binding pocket that is exposed to the cytosol (Fig. 6a, b)[23], suggesting diverse NADPH recognition mechanisms for membrane-embedded steroid reductases. We speculate that these enzymes are likely to adopt distinct structural

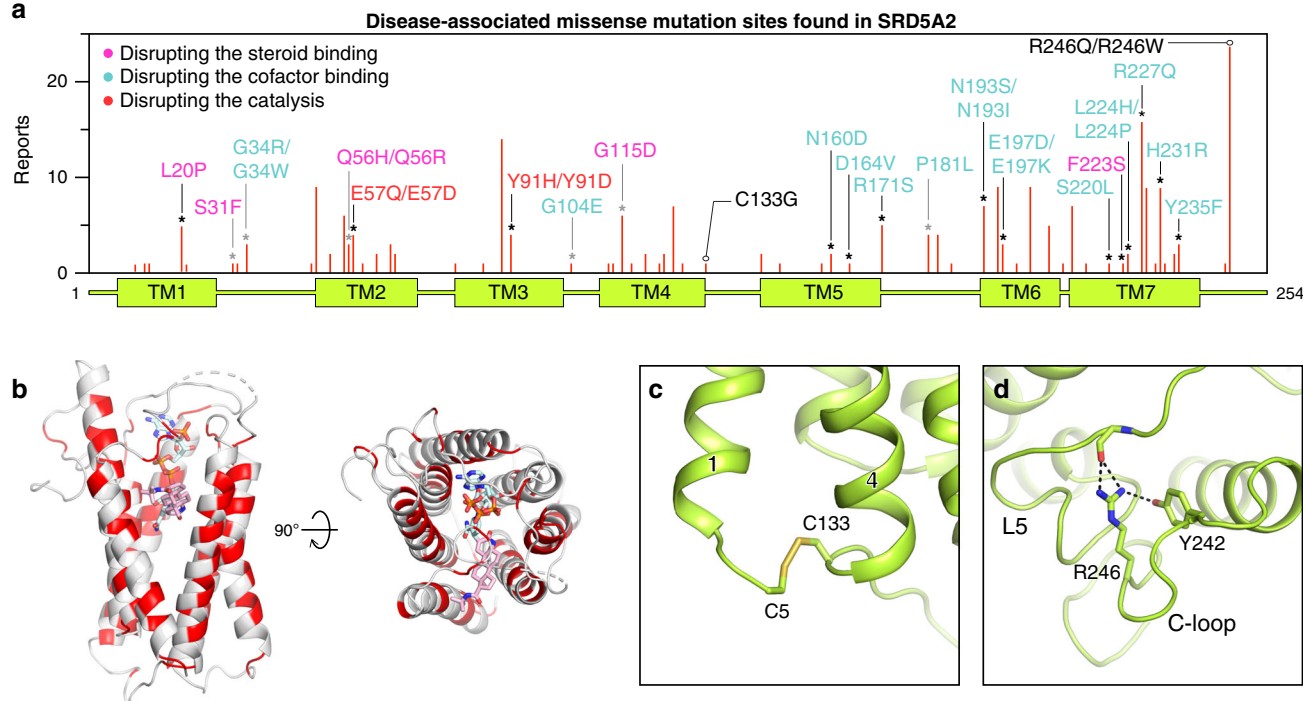

**Fig. 5 Structural analysis of disease-associated missense mutations of SRD5A2. a** Distribution of identified missense mutations of SRD5A2. The red bar length for each mutation site indicates how many times it was reported based on the data collected from HGMD and literature. The sites that are involved in the binding of NADP–DHF and the formation of the binding cavity are indicated by black and gray asterisks, respectively. The mutations presumably disrupting the steroid binding, the cofactor binding, and the catalysis are labeled in pink, cyan, and red, respectively. **b** Mapping of the mutation sites onto the SRD5A2 structure viewed from the side and the top of the transmembrane domain. The mutation sites are colored in red. **c, d** Environment of C133 and R246 suggesting their roles in protein folding.

features within their TM bundles, so that NADPH can be appropriately positioned toward their steroid substrates for site-specific carbon–carbon double bond reduction. Further structural investigation of other important eukaryotic membrane-embedded steroid reductases, including SRD5A3 and DHCR7, is needed to understand the potentially different mechanisms by which these enzymes reduce specific carbon–carbon double bonds at different positions of steroid or lipid substrates, using NADPH. Interestingly, SRD5A2 also shares a certain structural similarity with another eukaryotic integral membrane enzyme, the isoprenylcysteine carboxyl methyltransferase (ICMT) from *Tribolium castaneum*[52]. In particular, TMs 3–7 of SRD5A2 can be well aligned to TMs 4–8 of ICMT; the binding site for NADP in SRD5A2 also partially overlaps with the binding site for the cofactor S-adenosyl-L-homocysteine (SAH) in ICMT (Fig. 6c). Such structural similarity may suggest an evolutionarily conserved feature in cofactor binding among these enzymes.

## Methods

**SRD5A2 expression and purification.** cDNA of human full-length wild-type SRD5A2 was synthesized (IDT) and cloned into pFastBac vector with an N-terminal signal peptide followed by a FLAG epitope and an 8× histidine tag. One tobacco-etch virus (TEV) protease cleavage was introduced after the His tag. SRD5A2 protein was expressed in the insect Sf9 cells using the Bac-to-Bac baculovirus system (Thermo-Fisher). Cells were infected by baculovirus at a density of $4.0 \times 10^6$ cells ml$^{-1}$ and harvested after 48 h at 27 °C. To stabilize the protein, all purification steps were accomplished in the presence of the inhibitor finasteride (Tocris). All buffers mentioned below were prepared by mixing each ingredient from stock solutions in the following order: buffering agents, salts, detergents (where applicable), ligand, protein inhibitors (where applicable), and other ingredients.

Pelleted Sf9 cells were lysed in lysis buffer containing 20 mM Tris pH7.5, 2.0 mg ml$^{-1}$ iodoacetamide, 0.2 μg ml$^{-1}$ leupeptin (Sigma), 100 μg ml$^{-1}$ benzamidine (Sigma), and 0.5 μM finasteride by stirring at 4 °C for 1 h. For cell pellets from 1 l cell media, 200 ml lysis buffer was used. The cell pellet was further resuspended and homogenized using the glass dounce homogenizer for 30–40 strokes on ice in solubilization buffer containing 20 mM HEPES pH7.5, 750 mM NaCl, 1.0 μM finasteride, 20% glycerol, 1% dodecyl maltoside (DDM; Anatrace), 0.1% cholesterol hemisuccinate (CHS; Sigma), 0.2% sodium cholate, 2.0 mg ml$^{-1}$ iodoacetamide, 0.2 μg ml$^{-1}$ leupeptin, 100 μg ml$^{-1}$ benzamidine, and salt active nuclease (ArcticZymes) and incubated at 4 °C for 90 min. For cell pellets from 1 l cell media, 150 ml solubilization buffer was used. After centrifugation at $65,000 \times g$ for 30 min, the supernatant was collected and incubated with 10 ml Ni-NTA resin (Cytiva) in the presence of 8 mM imidazole to prevent the nonspecific binding at 4 °C overnight. The resin was pelleted by centrifugation at $3000 \times g$ for 10 min and washed with 100 ml buffer containing 20 mM HEPES pH7.5, 500 mM NaCl, 1.0 μM finasteride, 0.1% DDM, 0.02% CHS, and 40 mM imidazole. The bound protein was eluted using the same buffer with 400 mM imidazole. The eluted protein was then loaded onto 5 ml anti-FLAG M1 antibody affinity column after supplementing with 2 mM CaCl$_2$, since the binding of FLAG to M1 antibody requires Ca$^{2+}$ (ref. [53]). The M1 antibody resin was washed with: 50 ml buffer 1 containing 20 mM HEPES pH7.5, 100 mM NaCl, 1.0 μM finasteride, 0.1% DDM, 0.02% CHS, and 0.1% lauryl maltose neopentyl glycol (LMNG; Anatrace) for 20 min; then 50 ml buffer 2 containing 20 mM HEPES pH7.5, 100 mM NaCl, 1.0 μM finasteride, 0.02% DDM, 0.02% CHS, and 0.1% LMNG for 20 min; and then 20 ml buffer 3 containing 20 mM HEPES pH7.5, 100 mM NaCl, 1.0 μM finasteride, 0.02% CHS, and 0.1% LMNG. This step was to extensively and slowly exchange detergent from 0.1% DDM to 0.1% LMNG. The protein was eluted from the M1 antibody resin using buffer containing 20 mM HEPES pH7.5, 100 mM NaCl, 1.0 μM finasteride, 0.01% LMNG, 0.001% CHS, 200 μg ml$^{-1}$ synthesized FLAG peptide (GL Biochem), and 5 mM EDTA. A total of 10 μl home-made TEV protease and 5 μl PNGase F (NEB) were added to the eluted protein and incubated at 4 °C overnight. TEV protease was used to cleave off the N-terminal FLAG tag. PNGase F was used to reduce potential protein N-linked glycosylation. The treated sample was concentrated using protein concentrators with a molecular-weight cutoff of 50 K Dolton by centrifugation to 500 μl. The concentrated protein was further purified by SEC using a Superdex S200 increase column (Cytiva) pre-equilibrated with running buffer containing 20 mM HEPES pH7.5, 100 mM NaCl, 1.0 μM finasteride, 0.01% LMNG, and 0.001% CHS. The monodispersed fractions were pooled together and concentrated using the same protein concentrators to 50 mg ml$^{-1}$ for crystallization.

For all the buffers mentioned above: Tris and HEPEs were used as the buffering agents. NaCl was used to keep appropriate ionic strength. DDM and sodium cholate were detergents used to solubilize cell membrane, and keep membrane

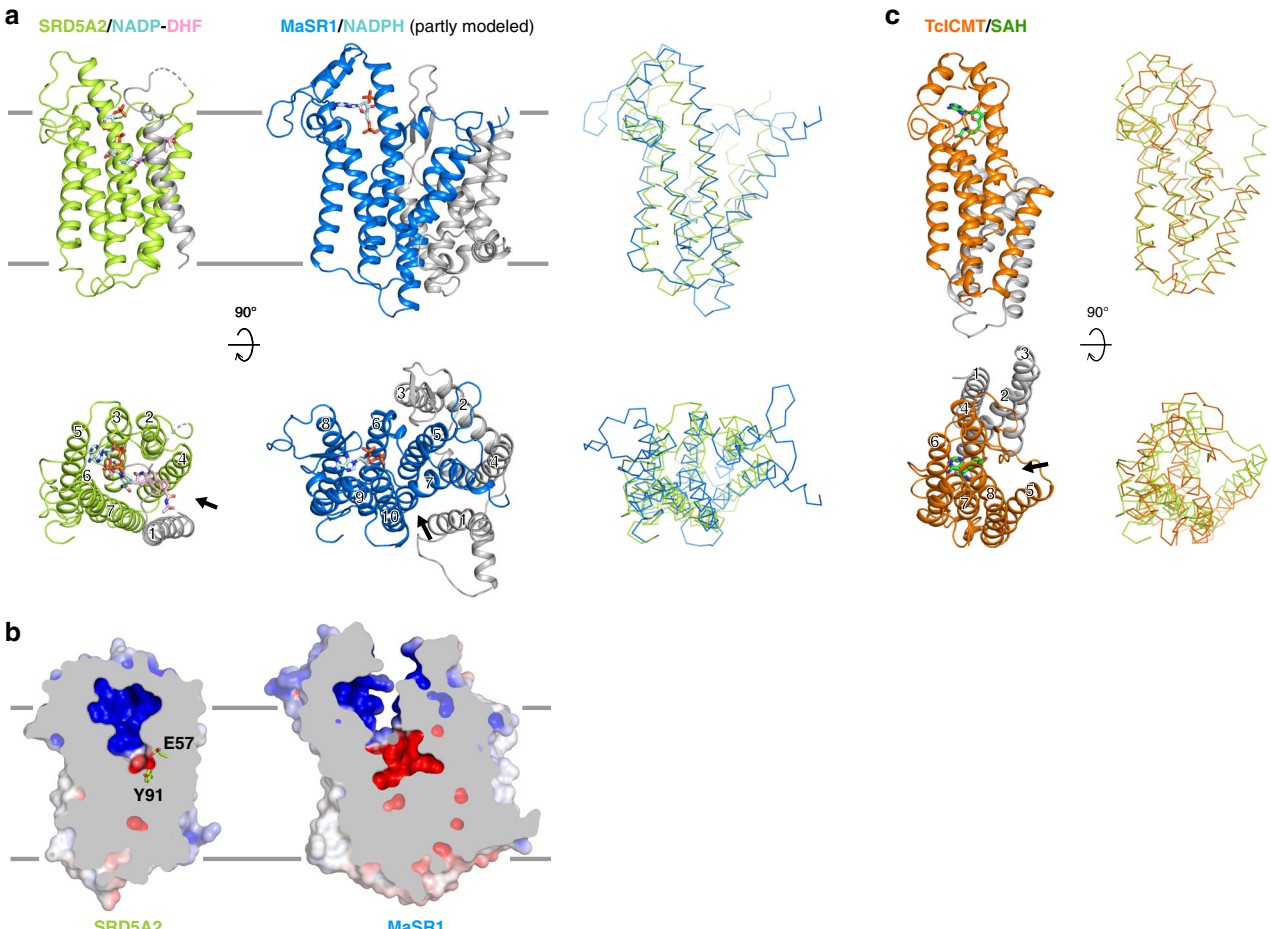

**Fig. 6 Structural comparison of SRD5A2 with MaSR1 and ICMT. a** Structural comparison of the transmembrane regions of SRD5A2 and MaSR1 (PDB ID: 4QUV). The core transmembrane regions in SRD5A2 and MaSR1 that can be structurally aligned are shown as colored ribbons, while the rest parts are shown as gray ribbons. The right panel shows the superimposition of SRD5A2 and MaSR1. **b** Occluded and exposed ligand-binding cavities in SRD5A2 and MaSR1, respectively. **c** Structural comparison of the transmembrane regions of SRD5A2 and ICMT (PDB ID: 5V7P). The core transmembrane region of ICMT that can be well aligned to SRD5A2 is colored orange. The cofactor of ICMT, SAH, is colored green. The right panel shows the superimposition of SRD5A2 and ICMT. The potential substrate entry ports in all three enzymes are indicated by arrows.

proteins soluble. Glycerol was used to stabilize the protein during solubilization. CHS was added in case the protein needs cholesterol to be stable. Iodoacetamide was used to prevent protein aggregation due to nonspecific disulfide bonds. Leupeptin and benzamidine were used as protease inhibitors. Salt active nuclease was used to degrade DNA to lower viscosity of solubilized materials.

**Crystallization**. Purified protein in complex with finasteride was reconstituted into the lipidic cubic phase (LCP)[28,29] through mixing protein with monoolein/cholesterol lipid mixture (10:1 w/w; Avanti) at a weight ratio of 2:3 (protein:lipid), using coupled syringes (ArtRobbins). Using a Gryphon LCP robot (ArtRobbins), the LCP mixture was dispensed onto 96-well glass sandwich plates with 3 M tape 9500CC spacers (Saunders; ~0.14 mm well depth) in 20 nl drops and then overlaid with 700 nl precipitant conditions. Plates were incubated at 20 °C and crystals grew in the condition of 100 mM tris-sodium citrate pH 5.0, 28–34% PEG600, 100–150 mM NaCl, and 100 mM $Li_2SO_4$. Crystals were collected from the LCP matrix and frozen in liquid nitrogen for data collection.

**De novo structure modeling for structure determination**. To predict the 3D model of human SRD5A2, similar to trRosetta[54], we used a de novo approach from a predicted distance/orientation matrix, which was derived from a variety of multiple sequence alignments, using three sequence databases, UniClust30 (UC)[55], UniRef90 (UR)[56], and NR[57], and then fused by implementing a deep residual network with a strip pooling module[58] to effectively capture long-range relationship of residual pairs. Following trRosetta[54], we generated the 300 3D models of SRD5A2 from the predicted distance and orientation, using constrained minimization. Specifically, the predicted distance and orientation probabilities were first converted into potentials, which are then used as restraints to be fed into Rosetta together with coarse-grained energy optimization. Finally, the top 1 structure satisfying the restraints was selected according to Rosetta energy, as the initial

model for structure determination. Details of our approach are also shown in Extended Data Fig. 3. We expect to incorporate our steps together as a web server, and make it available to the public in the near future.

**X-ray data collection and structure determination**. X-ray diffraction data were collected at the beamline 23ID-B, GM/CA of Advanced Photon Source (APS) in the Argonne National Laboratory at Chicago. Crystals were in a liquid nitrogen stream during data collection. Each crystal was exposed with a 10 μm× 10 μm beam at a wavelength of 1 Å for 0.2 s, and 0.2-degree oscillation per frame to collect 20 degrees of rotation data. Data sets from five crystals were processed and merged in HKL2000 software[59].

To determine the phase of the crystal structure, the de novo structure model was used as the search model for molecular replacement by Molrep in CCP4 package[60]. The initial phase was largely improved by using density modification methods in PHENIX[61]. COOT[62] was used for model rebuilding based on the improved electron density. The rebuilt model was further refined in PHENIX with an additional TLS refinement was performed. The model quality was check by MolProbity[63]. Ramachandran analysis showed that 95.9% residues are in the favored region and 4.1% residues are in the allowed region. The final refinement statistics are listed in Table 1. All structure figures were prepared by PyMOL (http://www.pymol.org/).

**Molecular docking**. The NADP–DHF adduct from the crystal structure was removed and an analogous NADP–testosterone adduct with NADP covalently bound to the C-5 atom of testosterone at the α-face was docked to the crystal structure. In the docked pose, the covalent link was removed to obtain NADPH and testosterone. This was followed by optimizing the ligands and their surrounding 5 Å protein residues. Thereafter, NADPH was redocked to the optimized protein, followed by redocking testosterone to the NADPH docked protein.

The docking poses were filtered to be within 10 Å RMSD of the X-ray ligand densities to obtain the pose for NADPH and testosterone docked to SRD5A2. Docking calculations were performed using Glide[64].

**Molecular modeling and MD simulations**. SRD5A2 models were built based on the solved crystal structure to incorporate unresolved residues 1–4 (N-terminus) and 39–43 (loop L1) using Modeller[65]. We generated 5000 models and selected top ten scoring models based on DOPE scoring function[66]. From the top ten models, two most structurally diverse protein configurations were selected to start MD simulations for unliganded (apo0 and apo1) and NADP$^+$-bound (nap0 and nap1) states. NADP$^+$ was placed at the NADP density in the crystal structure. NADP$^+$ was modeled at a molecular charge of −2. Ionization states of protein residues were calculated at pH = 7 by solving Poisson–Boltzmann equation in continuum electrostatics models ($\varepsilon_{\text{protein}} = 20$; $\varepsilon_{\text{solvent}} = 80$) using APBS[67]. All residues were found to be in their standard ionization states. Based on the crystal structure, a disulfide bond was modeled between Cys-5 and Cys-133. The protein models were embedded in a pre-equilibrated 1-palmitoyl-2-oleoyl-sn-glycero-3-phosphocholine lipid bilayer followed by solvation (TIP3P water model) and neutralization, using potassium and chloride ions at 150 mM. The simulation setups comprised ca. 44,000 atoms each. CHARMM36 forcefield was employed for the MD simulations, including protein, ligands, lipids, water, and ions[68]. The systems were first energy minimized for 10,000 steps and then heated gradually from 0 to 310 K for 250 ps, using a Langevin thermostat with heavy atoms restrained at 10 kcal mol$^{-1}$ Å$^{-2}$ in an NVT ensemble. The heated systems were subjected to eight successive rounds of 1 ns equilibration steps. During the equilibration, protein and ligand heavy atoms were subjected to harmonic restraints, and lipids were subjected to planar restraints to maintain bilayer planarity. The harmonic restraints for each step were relaxed progressively going from 10 to 0.1 kcal mol$^{-1}$ Å$^{-2}$. The equilibrations were performed at a 1 fs timestep at $T = 310$ K and $P = 1$ bar using the Langevin thermostat and Nosé–Hoover Langevin barostat in NPT ensemble. The production runs were performed with a hydrogen mass repartitioning scheme with a timestep of 3.6 fs with a nonbonded cutoff at 12 Å (ref. [69]). Long-range electrostatics were evaluated with the particle mesh Ewald method. Protein and lipid bond lengths were constrained with the SHAKE algorithm. Each system was simulated for ca. 1.25 μs, giving a total simulation time of ca. 5.4 μs (2.7 μs apo and 2.7 μs nap). Trajectory snapshots were saved at every 50 ps. The simulations were performed with NAMD 2.13 (ref. [70]). The simulation setup was constructed using CHAMM-GUI[71]. In order to analyze the multidimensional conformational landscape, we performed PCA of MD trajectories to identify the dominant modes of protein motions (principal components) in the apo and nap simulation states. PCA was performed with pyPcazip[72]. Visual molecular dynamics (VMD)[73] was employed for visualization and for performing RMSF analysis to probe protein mobility.

**Enzyme activity measurement**. We used insect cell membranes overexpressing different constructs of SRD5A2 in the measurement of enzyme activity. Sf9 cells expressing wild-type SRD5A2 (WT), and mutants E57Q and Y91F were harvested after 48 h transfection. The expression level of each construct was determined by flow cytometry using FITC-labeled anti-FLAG M2 antibody (Sigma Aldrich) in the presence of 0.5% Triton X-100. For membrane preparation, the cell pellets were resuspended in lysis buffer containing 20 mM Tris pH7.5, 1 mM EDTA, 0.2 μg ml$^{-1}$ leupeptin, and 100 μg ml$^{-1}$ benzamidine. The lysed and homogenized samples were centrifuged at $1200 \times g$ for 8 min. The supernatant was further centrifuged at $300,000 \times g$ for 50 min to get the membrane pellets, which were resuspended in buffer containing 20 mM HEPES pH7.5, 100 mM NaCl, 1 mM EDTA, 0.2 μg ml$^{-1}$ leupeptin, and 100 μg ml$^{-1}$ benzamidine and stored at −80 °C.

The enzyme activity was assayed using the prepared membrane fractions (0.3 mg total protein determined by Bradford assay for each reaction) in the presence of 0.5 mM NADPH and 0.5 mM testosterone in 0.1 M sodium citrate pH 5.0 buffer. For the negative control, finasteride at 0.5 mM was added together with testosterone. After incubation for 4 h at 37 °C, the steroids were extracted with chloroform, evaporated by centrifugal concentrator, and re-dissolved in methanol for liquid chromatography–mass spectrometry (LC–MS) analysis, using buffer A containing 1 mM ammonium formate and buffer B containing 100% methanol. The areas of peaks corresponding to DHT and testosterone were integrated. The ratios of the peak area of DHT to the peak area of testosterone were calculated as indicators of enzyme activity. The data were processed by Prism8 (GraphPad). The experiments on the wt SRD5A2 (WT) were repeated five times. The experiments on WT with 0.5 mM finasteride were repeated five times. The experiments on each mutant, E57Q or Y91F, were repeated three times. The data were represented as mean ± SEM.

**Reporting summary**. Further information on research design is available in the Nature Research Reporting Summary linked to this article.

## Data availability
Atomic coordinates for the structure of SRD5A2 have been deposited in the world-wide Protein Data Bank (wwPDB) (https://www.wwpdb.org/) under the accession numbers 7BW1. Protein structures with PDB IDs: 4QUV and 5V7P have been referenced in the manuscript. All relevant data are available from the authors. Source data are provided with this paper.

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

## Acknowledgements

We thank the staff of the GM/CA of APS at the Argonne National Laboratory in Chicago for assistance with X-ray diffraction data collection, and Southern University of Science and Technology Core Research Facilities for assistance with LC–MS analysis. This work was supported by the National Institutes of Health (NIH) grants R35GM128641 (to C.Z.), the National Natural Science Foundation of China 31971131 and 31770791, Shenzhen-Hong Kong Institute of Brain Science, Shenzhen Fundamental Research Institutions 2019SHIBS0002 (to Z.W.), and funding support from the Biomedical Research Council of A*STAR (to S.S. and H.F.). The molecular dynamics simulations were performed on resources of the National Supercomputing Centre, Singapore (https://www.nscc.sg).

## Author contributions

C.Z. and Z.W. supervised the project and designed the research with H.F., L.W., and Q.X. Q.X. performed the experiments for protein expression, purification, and crystallization with the assistance from L.W. L.W. performed X-ray diffraction data collection with the assistance from H.L. C.Z. processed the X-ray diffraction data. T.S., F.Y., and J.H. performed de novo structure modeling. Z.W. determined the crystal structure. S.S. and H.F. performed computational docking and simulation studies. C.Z., Z.W., S.S., and H.F. prepared the manuscript with the assistance from Q.X.

## Competing interests

The authors declare no competing interests.
