## [Peer Review File · Nature Communications]

REVIEWER COMMENTS

Reviewer #1 (Remarks to the Author):

The manuscript by Zhang and colleagues addresses the questions of how steroid 5 α -reductase 2 (SRD5 α 2) catalyzes testosterone to dihydrotestosterone and how finasteride inhibits the reductase activity. This study answers several important questions for steroid metabolism that will be of broad interest. SRD5 α 2 is a 7-TMs transmembrane enzyme that catalyzes the irreversible reduction of the Δ 4,5 bond in Δ 4-3-ketosteroids using reduced reagent NADPH as the hydride donor co-factor. The authors use X-ray crystallography to determine the structure of SRD5 α 2 with its inhibitor finasteride and co-factor NADPH. Notably, finasteride has been shown to play an important role in reversing hair loss and it has been approved by FDA for clinical treatment. The structure is determined at atomic resolution and the parameters after the refinement reveal this structure is determined with high quality. This referee has checked the electronic map and structure mode, and has no significant concern regarding the structural data. Overall, this manuscript should be suitable to publish in a high impact journal, but several points/concerns regarding the presentation and discussion of this study should be clarified or addressed before acceptance.

1. Structural determination (Page 4): The authors claimed that "The structure was determined to 2.8-Å resolution by molecular replacement based on a structural model from de novo prediction since we failed to use anomalous diffraction data to solve the structure (see Methods)." This is a quite interesting point that authors could use the predicted model for molecular replacement (MR) to determine the structure. After carefully reading the method, this referee thinks that the authors should make a supplementary figure to show the details in the model prediction and MR procedure. In this figure the authors should show what protein sequence they used for the alignment, how many models they generated, how they optimized these models using Rosetta, and the Z-score for the individual model after MR. This referee thinks that this technique is really remarkable, and the X-ray crystallography community will greatly benefit from a detailed protocol.

2. The formation of NADP-DHF adduct (Page 5, figure 2): Although this covalent adduct has been proposed over twenty years ago, it is very exciting to see the electron density of this adduct is shown at atomic resolution in this manuscript. However, several points about the formation of this adduct should be discussed.

- a. Where is the energy for the reaction? Is water involved in the reaction?
- b. Is this reaction reversible? Will NADP-DHF lead to a permanent inhibition of SRD5 α 2?
- c. How about the inhibitory mechanism of other SRD5 α 2 inhibitors?
- d. The source data of Fig. 3e should be provided in the revision.

3. Structural presentation: The authors claimed, "The mutations presumably disrupting the steroid binding, the cofactor binding, and the catalysis are labeled in pink, cyan and red respectively." in the figure legend. This referee suggests authors to indicate these colors in the figure. Also, Fig. 5b is very difficult to follow, the authors should show another orientation (parallel to the membrane) as well.

4. Structural comparison and related discussion:

- a. The maSR1 shares very high structural similarity with methyltransferase ICMT (isoprenylcysteine carboxyl methyltransferase) (Yang et al Mol Cell 2011 and Li et al Nature 2015). Notably, the eukaryotic intramembrane Ras methyltransferase ICMT has been determined in 2018 (Diver et al Nature 2018). This referee has compared the structure of SRD5 α 2 with ICMT, it is very clear that the TMs 3-7 share a similar fold with TMs 4-8 of ICMT; particularly, the adenine ring of SAH in ICMT share a similar position of adenine ring of NADP in SRD5 α 2. It seems to be an evolutionarily conserved feature among these enzymes as well as steroid reductases, sterol reductases, and methyltransferase. The authors should compare the structure with ICMT (pdb: 5V7P) and discuss this point in the main text.
- b. The SRD5 α 2 is a 7-TMs enzyme, albeit the topology of this enzyme is distinct from GPCR. Some

GPCRs (SMO, bile acid receptor) have been shown to accommodate sterol molecules by its 7-TMs. It would be beneficial to mention/compare the sterol binding characters of these receptors with SRD5a2.

5. Map presentation: The authors should show 2Fo-Fc map of each helix as well as the map of disulfide bond in a supplementary figure.

Reviewer #2 (Remarks to the Author):

Human steroid 5 α -reductases (SRD5a α s) mainly catalyze the irreversible reduction of the Δ 4,5 bond in Δ 4-3-ketosteroids using NADPH as a co-factor. SRD5a2 is the most intensively investigated SRD5a which catalyzes testosterone to dihydrotestosterone (DHT), the major steroid hormone for androgen receptor. Abnormal activities of SRD5a2 result in diseases such as benign prostatic hyperplasia, androgenic alopecia, prostatic cancer and male infertility. Finasteride and dutasteride as SRD5a2 inhibitors are widely used antiandrogen drugs. Recently, androgen receptor signaling has been linked to COVID-19 disease severity, suggesting the repurposing potential of SRD5a2 inhibitors for the treatment of COVID-19. However, the molecular mechanisms underlying enzyme catalysis and inhibition remained elusive for SRD5a2 and other eukaryotic integral membrane steroid reductases due to a lack of structural information. In this paper, the authors solved a structure of human SRD5a2 in the presence of NADPH and finasteride. The authors proposed molecular mechanisms for catalytic mechanism of SRD5a2 and irreversible inhibition of finasteride based on structural analysis and mutagenesis studies. MD simulations were also performed on microsecond timescales to investigate the conformational dynamics of SRD5a2. Dramatic conformational changes of loop L1 were revealed, and the authors proposed that the conformational changes allow the NADP⁺/NADPH exchange during the reaction. The manuscript is clearly written and well organized. The methods and results are rigorous. The statistical analysis is appropriate. It is recommended for publication after the following questions are addressed.

Minor concerns:

1. On Page 15, in the 'De novo structure modeling' section, the authors mentioned an approach of generating the initial models for structure determination. I'm wondering if the same approach has been applied previously, and if this approach generates initial models with better quality than other methods.
2. Please indicate in the main text whether the authors ever attempted to solve the structure of human SRD5a2 in the absence of finasteride, and if yes, explain briefly why the structure of SRD5a2 in complex with NADPH cannot be obtained.
3. Page 7: "In the docked structure, while the NADPH molecule could be well aligned to the NADP moiety of NADP-DHF in the crystal structure ..." Please specify the RMSD of NADPH between the docked structure and the crystal structure.
4. Figure caption of Fig. 3: "The finasteride and NADPH conformations in the AKR1D1 structure (PDB ID 3G1R) were shown by align ..." should be "... (PDB ID: 3G1R) were shown by aligning ..." Also, please also specify the distances of the two hydrogen bonds in Fig. 3C.

Reviewer #3 (Remarks to the Author):

The structure of steroid 5 α -reductase type 2 (SRD5A2) determined by X-Ray crystallography is reported. The data show that this membrane-bound enzyme adopts a structure containing seven transmembrane domains connected by six loops that extend into the cytosol or endoplasmic

reticulum. The transmembrane domains are predicted to enclose an active site in which a steroid substrate and NADPH cofactor interact. The structure explains many of the previously determined enzymatic and kinetic properties of the enzyme and differences with other steroid reductases in these parameters. It also confirms that 4-azasteroids with $\Delta 1,2$ configurations like finasteride and dutasteride inhibit the enzyme by driving the formation of a covalent adduct between the NADPH-cofactor and the inhibitor.

SRD5A2 plays a major role in formation of the male phenotype and is a target for the treatment of benign prostatic hyperplasia and prostate cancer, thus the structure of this enzyme represents an important advance and will be of broad interest to endocrinologists, urologists, human geneticists, big pharma, and biochemists.

Major Comments

1. Efforts to solubilize membrane-bound steroid 5alpha-reductase activity with detergents have met with failure in the past and thus the authors are to be commended for deducing the combination of detergents and purification procedures that led to the solved structure. This being written, there are no data in the paper to indicate that the purified protein used for crystallization is enzymatically active. Membrane pellets from insect cells expressing the baculovirus-SRD5A2 vector are shown in Figure 3e to have 5alpha-reductase activity in the absence of detergents; however, one cannot assume that the same results would be obtained with protein purified in the presence of multiple detergents. To ensure that the determined structure represents that of an active enzyme, the authors should add data showing that the purified protein is able to convert testosterone to dihydrotestosterone with a reasonable specific activity.
2. Sequence conservation and/or divergence in the active sites of the SRD5A1 and SRD5A2 proteins may, as the authors illustrate in Extended Data Figure 4f, explain why finasteride is ~100-times more potent at inhibiting SRD5A2 versus SRD5A1. Does the sequence conservation suggest why dutasteride is equally potent at inhibiting both SRD5A1 and SRD5A2? Also, do the sequences suggest why the two enzymes have radically different pH optima? SRD5A2 has an acidic pH optimum of 5.0-5.5 in in vitro enzyme assays, whereas SRD5A1 has a basic pH optimum of >8.0. In the early days of steroid 5alpha-reductase research, the existence of two isozymes was first postulated based on these different pH optima.
3. The work of Makridakis et al. (references 28, 29, and 39 in the present paper) has never been reproduced and thus the contention that gain-of-function mutations exist in SRD5A2 and predispose to prostate cancer is contentious. In the absence of showing enzymatic data that confirm the A49T and A248V alterations are more active, it might be best not to refer to these papers.

Minor Comments

1. The correct symbols for the steroid 5alpha-reductase 1 and 2 proteins, respectively are SRD5A1 and SRD5A2, not SRD5a1 and SRD5a2.
2. Figure 1 legend: "...different R-groups connected in the tails." should read "...different R-groups connected to the amide side chains."
3. Page 6, "...explaining the selectivity of finasteride for human SRD5a1 over SRD5a2." should read "explaining the selectivity of finasteride for human SRD5A2 over SRD5A1".

Reviewer #4 (Remarks to the Author):

The manuscript describes a 2.8 Å crystal structure of a human membrane steroid reductase from the ER responsible for the NADPH dependent reduction of testosterone to dihydrotestosterone. The structure was obtained as a complex with the dualsteric ligand NADP-dihydrofinasteride and provides insight into the manner in which substrates bind the enzyme and the reaction and inhibition mechanisms. Disease causing mutations that raise and lower enzyme activity and ligand selectivity are rationalized based on the structure. The work provides new insights into the mechanism of action and inactivation of a medically important enzyme. Structures of a related bacterial membrane and of a soluble variant of the enzyme are known. While there are similarities, the differences highlighted in

the present work represents an important advance for the field. The conclusions and claims of the study are supported by convincing data. The methodology is sound and is of a high standard. However, not enough detail is provided in the methods for the work to be reproduced. The English composition and grammar will need correction.

Points that warrant attention follow.

Discuss the reasons why there are no apo-structures and structures of complexes with one or other substrate? Relate this to the kinetic mechanism of the reaction if known. Is it ping-pong or other, for example?

25 The binding cavity is not 'inside the membrane'

28 Do the simulations tell us anymore that what is already available from crystallographic B-factors and missing residues?

38 List the other lipid substrates.

41 Explain how these enzymes are involved in N-linked glycosylation.

46 inefficient/insufficient?

56 The link to Covid-19 is not clear and should be explained.

79 The missing residues in L1 need to be referred to in the simulations work.

80 Explain dualsteric for the benefit of the general reader.

98 Shorter than 2 Å is not useful. State actual bond length.

Fig. 2 Show the entire molecule in b to match that in a.

Show membrane boundaries in c.

State criteria for a hydrogen bond.

110 Provide values for binding constants and off rates.

189 Use the same color scheme for the two NADPs and finasterides in f

198 Explain why finasteride does not inhibit 100% in e.

218 Has the pKa of E57 been estimated in support of the assumed protonation at pH~5?

219 could/does

231 It is important to alert the reader to the fact that the crystal structure is missing 5 residues in L1 and that these have been modelled back in for simulations work.

Fig. 6 The superpose is not obvious in a.

Show membrane boundaries in a and b

Table 1. Three decimal places are not justified for cell dimensions.

Methods section must be rewritten with enough detail for the work to be reproduced. Concentrations, volumes, order of addition, mixing, time, temperatures, numbers of dounce homogenizations, etc must be reported. Reasons for including various additives (IAA, benzamidine, calcium, etc) should be explained.

478 How was the protein concentrated? What is the composition of the final buffer?

480 LCP reference is incorrect. It should be the Nature Protocols 2009 reference.

483 What spacer thickness was used?

527 Was E57 in its standard state. Indicate what the standard state is and how this relates to the proposed mechanism.

561 mg of what? How was it quantified?

563 Explain how the negative control was run.

566 Peak area of what? What signal was quantified? Is the response the same for the two analytes?

ED Fig 4 d and e switched.

State program used for 2D view.

Is there any significance attached to the difference seen between the WT and the two mutants in expression levels in h?

Title: **Structure of human steroid 5 α -reductase 2 with anti-androgen drug finasteride**

We thank the reviewer for the constructive comments. Please see our detailed responses to the comments below. The reviewer's comments are in **blue** font and our responses are in **black** font.

Reviewer #1:

The manuscript by Zhang and colleagues addresses the questions of how steroid 5 α -reductase 2 (SRD5 α 2) catalyzes testosterone to dihydrotestosterone and how finasteride inhibits the reductase activity. This study answers several important questions for steroid metabolism that will be of broad interest. SRD5 α 2 is a 7-TMs transmembrane enzyme that catalyzes the irreversible reduction of the Δ 4,5 bond in Δ 4-3-ketosteroids using reduced reagent NADPH as the hydride donor co-factor. The authors use X-ray crystallography to determine the structure of SRD5 α 2 with its inhibitor finasteride and co-factor NADPH. Notably, finasteride has been shown to play an important role in reversing hair loss and it has been approved by FDA for clinical treatment. The structure is determined at atomic resolution and the parameters after the refinement reveal this structure is determined with high quality. This referee has checked the electronic map and structure mode, and has no significant concern regarding the structural data. Overall, this manuscript should be suitable to publish in a high impact journal, but several points/concerns regarding the presentation and discussion of this study should be clarified or addressed before acceptance.

1. Structural determination (Page 4): The authors claimed that "The structure was determined to 2.8-Å resolution by molecular replacement based on a structural model from de novo prediction since we failed to use anomalous diffraction data to solve the structure (see Methods)." This is a quite interesting point that authors could use the predicted model for molecular replace (MR) to determine the structure. After carefully reading the method, this referee thinks that the authors should make a supplementary figure to show the details in the model prediction and MR procedure. In this figure the authors should show what protein sequence they used for the alignment, how many models they generated, how they optimized these models using Rosetta, and the Z-score for the individual model after MR. This referee thinks that this technique is really remarkable, and the X-ray crystallography community will greatly benefit from a detailed protocol.

We thank the reviewer for pointing out the significance of our approach. We have included a new **Extended Data Figure 3** showing the details of structure modeling and MR process and also revised our Methods to include the details of our approach. We also cited a previous study using a similar approach in structure determination¹ in our revised manuscript.

In fact, we are strenuously developing our *de novo* structure modeling approach to make it a better tool for the general scientific community. For example, since late-2019, we have registered our approach as Server83 to Server89 in CAMEO (Continuous Automated Model Evaluation)², a world-leading automatic structure prediction evaluation platform which includes Robetta³ from Baker group, HHpred⁴ from Soeding group, RaptorX⁵ from Xu group, SwissModel⁶ from Torsten group, etc. It is worth mentioning that since June 2020, our registered servers have been the weekly, monthly, and quarterly champion teams till now. The IDDT (local Distance

Difference Test)⁷ scores from the best of our servers on those targets are 59.1 (from 2020-05-08 to 2020-08-01), which greatly surpassed Robetta (IDDT value is 47.7). In the near future, we shall incorporate all details of our approach as a web server and make it available to the public.

2. The formation of NADP-DHF adduct (Page 5, figure 2): Although this covalent adduct has been proposed over twenty years ago, it is very exciting to see the electron density of this adduct is shown at atomic resolution in this manuscript. However, several points about the formation of this adduct should be discussed.

a. Where is the energy for the reaction? Is water involved in the reaction?

For most chemical reactions, the source of the activation energy is usually heat energy from the environment. There is energy coupling for some cellular reactions in a way that the free energy released from one reaction can be absorbed by another reaction usually with ATP as the mediator. For the reduction of steroid hormones in the cells, to the best of our knowledge, there is no energy coupling with other reactions. We believe this reaction happens simultaneously driven by environmental heat energy and SRD5A2 acts as the catalyst to lower the activation energy to facilitate and speed up the reaction. Also, it is likely that the adduct formation between NADPH and finasteride is a thermodynamically favored reaction.

As for the water involvement, we didn't observe clear electron density for well-ordered water molecules in the ligand-binding pocket. It could be due to the limitation of resolution of our structure or the actual absence of water molecules during the reaction. We cannot rule out the possibility that a water molecule may serve as another proton donor for the reduction of testosterone. However, so far there was no evidence supporting it. This is also the case for the soluble 5 β -reductase AKR1D1. Even though its ligand-binding cavity is highly exposed to solvent, multiple high-resolution structures of AKR1D1 bound to different substrates didn't suggest the involvement of water molecules in its catalysis^{8,9}.

b. Is this reaction reversible? Will NADP-DHF lead to a permanent inhibition of SRD5 α 2?

The NADP-DHF adduct is unstable in its free form¹⁰. It can decompose to DHF (dihydrofinasteride)¹⁰. However, it forms very stable complex with the enzyme SRD5A2. The dissociation (release of DHF from the enzyme) constant K_{off} has been determined to be $\sim 2.57 \times 10^{-7} \text{ s}^{-1}$, which means the half-life of the enzyme-inhibitor complex is about 31 days and it is virtually irreversible¹⁰. In fact, as mentioned in the section 'Intermediate adduct formed between finasteride and NADP', NADP-DHF adduct has been suggested to rank among the most potent non-covalent enzyme inhibitors in general^{10,11}.

c. How about the inhibitory mechanism of other SRD5 α 2 inhibitors?

There are two SRD5A2 inhibitors currently used as drugs, finasteride and dutasteride, which share the same sterol four-ring core structure (**Fig. 1b**). We believe the inhibitory mechanism is conserved for these two inhibitors. Indeed, the action of dutasteride has been shown to be irreversible as well¹². For other SRD5A2 inhibitors with different chemical scaffolds, which are much less studied than finasteride and dutasteride¹³, they may serve as the competitive inhibitors by simply occupying the testosterone binding pocket.

d. The source data of Fig. 3e should be provided in the revision.

We have submitted the source data together with the revised manuscript in the Excel file named S5A2_MS.

3. Structural presentation: The authors claimed, “The mutations presumably disrupting the steroid binding, the cofactor binding, and the catalysis are labeled in pink, cyan and red respectively.” in the figure legend. This referee suggests authors to indicate these colors in the figure. Also, Fig.5b is very difficult to follow, the authors should show another orientation (parallel to the membrane) as well.

We thank the reviewer for the suggestion. We have revised the **Figure 5** as suggested by this reviewer and Reviewer #3.

4. Structural comparison and related discussion:

a. The maSR1 shares very high structural similarity with methyltransferase ICMT (isoprenylcysteine carboxyl methyltransferase) (Yang et al Mol Cell 2011 and Li et al Nature 2015). Notably, the eukaryotic intramembrane Ras methyltransferase ICMT has been determined in 2018 (Diver et al Nature 2018). This referee has compared the structure of SRD5 α 2 with ICMT, it is very clear that the TMs 3-7 share a similar fold with TMs 4-8 of ICMT; particularly, the adenine ring of SAH in ICMT share a similar position of adenine ring of NADP in SRD5 α 2. It seems to be an evolutionarily conserved feature among these enzymes as well as steroid reductases, sterol reductases, and methyltransferase. The authors should compare the structure with ICMT (pdb: 5V7P) and discuss this point in the main text.

b. The SRD5 α 2 is a 7-TMs enzyme, albeit the topology of this enzyme is distinct from GPCR. Some GPCRs (SMO, bile acid receptor) have been shown to accommodate sterol molecules by its 7-TMs. It would be beneficial to mention/compare the sterol binding characters of these receptors with SRD5 α 2.

We thank the reviewer for the suggestion.

a. We have included structural comparison between SRD5A2 and ICMT at the end of the 'Discussion' section as suggested by the reviewer and added a new panel in **Figure 6 (Fig. 6c)**.

b. We have included discussion of different sterol ligand binding modes in GPCRs and in SRD5A2 in the second paragraph of section 'Binding pockets for NADP-DHF'. The sterol ligands in two GPCRs, SMO and the bile acid receptor GPBAR, adopt binding poses that are perpendicular to the membrane, contrasting the binding pose of DHF that is almost parallel to the membrane. We didn't observe obvious conserved features in sterol recognition, which may suggest highly diverse mechanisms of sterol ligand recognition by membrane proteins.

5. Map presentation: The authors should show 2Fo-Fc map of each helix as well as the map of disulfide bond in a supplementary figure.

We have included those maps in **Extended Data Figure 2d** as suggested by the reviewer.

Reviewer #2:

Human steroid 5 α -reductases (SRD5 α s) mainly catalyze the irreversible reduction of the Δ 4,5 bond in Δ 4-3-ketosteroids using NADPH as a co-factor. SRD5 α 2 is the most intensively investigated SRD5 α which catalyzes testosterone to dihydrotestosterone (DHT), the major steroid hormone for androgen receptor. Abnormal activities of SRD5 α 2 result in diseases such as benign prostatic hyperplasia, androgenic alopecia, prostatic cancer and male infertility.

Finasteride and dutasteride as SRD5 α 2 inhibitors are widely used antiandrogen drugs. Recently, androgen receptor signaling has been linked to COVID-19 disease severity, suggesting the repurposing potential of SRD5 α 2 inhibitors for the treatment of COVID-19. However, the molecular mechanisms underlying enzyme catalysis and inhibition remained elusive for SRD5 α 2 and other eukaryotic integral membrane steroid reductases due to a lack of structural information. In this paper, the authors solved a structure of human SRD5 α 2 in the presence of NADPH and finasteride. The authors proposed molecular mechanisms for catalytic mechanism of SRD5 α 2 and irreversible inhibition of finasteride based on structural analysis and mutagenesis studies. MD simulations were also performed on microsecond timescales to investigate the conformational dynamics of SRD5 α 2. Dramatic conformational changes of loop L1 were revealed, and the authors proposed that the conformational changes allow the NADP⁺/NADPH exchange during the reaction. The manuscript is clearly written and well organized. The methods and results are rigorous. The statistical analysis is appropriate. It is recommended for publication after the following questions are addressed.

Minor concerns:

1. On Page 15, in the ‘De novo structure modeling’ section, the authors mentioned an approach of generating the initial models for structure determination. I’m wondering if the same approach has been applied previously, and if this approach generates initial models with better quality than other methods.

As also suggested by Reviewer 1, we have shown the details of our approach in **Extended Data Figure 3** and revised our Methods accordingly. We have not tried other methods to generate initial models. A similar approach has been used previously in determining the crystal structure of bacterial peptidoglycan polymerase RodA¹. We have cited this study in our revised manuscript.

We are strenuously developing our *de novo* structure modeling approach to make it a better tool for the general scientific community. For example, since late-2019, we have registered our approach as Server83 to Server89 in CAMEO (Continuous Automated Model Evaluation)², a world-leading automatic structure prediction evaluation platform which includes Robetta³ from Baker group, HHpred⁴ from Soeding group, RaptorX⁵ from Xu group, SwissModel⁶ from Torsten group, etc. It is worth mentioning that since June 2020, our registered servers have been the weekly, monthly, and quarterly champion teams till now. The IDDT (local Distance Difference Test)⁷ scores from the best of our servers on those targets are 59.1 (from 2020-05-08 to 2020-08-01), which greatly surpassed Robetta (IDDT value is 47.7). In the near future, we shall incorporate all details of our approach as a web server and make it available to the public.

2. Please indicate in the main text whether the authors ever attempted to solve the structure of human SRD5 α 2 in the absence of finasteride, and if yes, explain briefly why the structure of SRD5 α 2 in complex with NADPH cannot be obtained.

We thank the reviewer for the suggestion. We have discussed our efforts of protein expression and purification with and without finasteride in the first paragraph of section 'Structure determination and overall structure of human SRD5A2'.

We tried to obtain structures of human SRD5A2 without finasteride, which was not successful. In our experiments, most of purified human SRD5A2 without finasteride existed as aggregates in

detergent buffers determined by size exclusion chromatography (see *Fig. R1* below). We believe finasteride or NADP-DHF can stabilize the protein to allow successful protein purification and crystallization. This is similar to G protein-coupled receptors (GPCRs), which also have large ligand-binding cavities inside the transmembrane domain. Ligand-dependent protein stabilization has been shown to be a fundamental property of GPCRs, and the use of appropriate ligands is critical to GPCR structural characterization¹⁴.

Figure R1. Purification of human SRD5A2 with or without finasteride by size exclusion chromatography using Superdex 200 Increase column.

The instability of purified SRD5A2 without ligand is also consistent with our MD simulation, which suggested a highly dynamic nature of unliganded SRD5A2 with highly flexible loop regions. NADPH itself may not be enough to stabilize the structure of SRD5A2.

3. Page 7: “In the docked structure, while the NADPH molecule could be well aligned to the NADP moiety of NADP-DHF in the crystal structure ...” Please specify the RMSD of NADPH between the docked structure and the crystal structure.

The RMSD of NADP between docked and crystal structure is 0.58 Å for heavy atoms. We have included this information in the revised manuscript.

4. Figure caption of Fig. 3: “The finasteride and NADPH conformations in the AKR1D1 structure (PDB ID 3G1R) were shown by align ...” should be “... (PDB ID: 3G1R) were shown by aligning ...” Also, please also specify the distances of the two hydrogen bonds in Fig. 3C.

We thank the reviewer for the suggestions. We have revised the figure caption and specified the distances of hydrogen bonds in the revised **Figure 3**.

Reviewer #3:

The structure of steroid 5 α -reductase type 2 (SRD5A2) determined by X-Ray crystallography is reported. The data show that this membrane-bound enzyme adopts a structure containing seven transmembrane domains connected by six loops that extend into the cytosol or endoplasmic reticulum. The transmembrane domains are predicted to enclose an active site in

which a steroid substrate and NADPH cofactor interact. The structure explains many of the previously determined enzymatic and kinetic properties of the enzyme and differences with other steroid reductases in these parameters. It also confirms that 4-azasteroids with $\Delta 1,2$ configurations like finasteride and dutasteride inhibit the enzyme by driving the formation of a covalent adduct between the NADPH-cofactor and the inhibitor.

SRD5A2 plays a major role in formation of the male phenotype and is a target for the treatment of benign prostatic hyperplasia and prostate cancer, thus the structure of this enzyme represents an important advance and will be of broad interest to endocrinologists, urologists, human geneticists, big pharma, and biochemists.

We thank the reviewer for the positive comments.

Major Comments

1. Efforts to solubilize membrane-bound steroid 5 α -reductase activity with detergents have met with failure in the past and thus the authors are to be commended for deducing the combination of detergents and purification procedures that led to the solved structure. This being written, there are no data in the paper to indicate that the purified protein used for crystallization is enzymatically active. Membrane pellets from insect cells expressing the baculovirus-SRD5A2 vector are shown in Figure 3e to have 5 α -reductase activity in the absence of detergents; however, one cannot assume that the same results would be obtained with protein purified in the presence of multiple detergents. To ensure that the determined structure represents that of an active enzyme, the authors should add data showing that the purified protein is able to convert testosterone to dihydrotestosterone with a reasonable specific activity.

As mentioned in our responses to Reviewer 2, we found in our previous experiments that if we solubilized insect cell membranes expressing human SRD5A2 without finasteride, most of the protein aggregated in detergent buffers during purification as indicated by size exclusion chromatography (see **Fig. R1** above in our responses to Reviewer 2), suggesting protein misfolding. This is consistent with the previous efforts mentioned by the reviewer. We believe finasteride or NADP-DHF can stabilize the protein to allow successful protein purification and crystallization. Indeed, with finasteride we could get purified human SRD5A2 in detergent buffers running as a single peak corresponding to a monomeric state in size exclusion chromatography (**Extended Data Fig. 2a**). Such ligand-dependent protein stabilization is not uncommon. In fact, it has been shown to be a fundamental property of G protein-coupled receptors (GPCRs)¹⁴, which also have large ligand-binding cavities inside the transmembrane domain. The use of appropriate ligands is critical to purification and structural characterization of GPCRs¹⁴.

Considering the extremely slow dissociating property of NADP-DHF¹⁰, it would be very difficult to measure the enzyme activity of purified SRD5A2 in detergent buffers with finasteride. Nevertheless, the monodisperse state of purified SRD5A2 with finasteride and the fact that it could be crystallized in lipidic cubic phase (LCP)¹⁵ highly imply that the protein structure was not significantly disturbed during protein purification. In addition, our structural insight into enzyme catalysis agrees with previously reported and our mutagenesis data (**Figs. 3e and 5a**).

Interestingly, a recent study showed that purified human SRD5A2 from *E. coli* in detergents was not catalytically active, but subsequent reconstitution into liposomes could rescue its activity¹⁶. Similarly, we purified human SRD5A2 with finasteride in detergent buffers and then reconstituted it in a lipidic environment (lipid mesophase) to obtain protein crystals.

2. Sequence conservation and/or divergence in the active sites of the SRD5A1 and SRD5A2 proteins may, as the authors illustrate in Extended Data Figure 4f, explain why finasteride is ~100-times more potent at inhibiting SRD5A2 versus SRD5A1. Does the sequence conservation suggest why dutasteride is equally potent at inhibiting both SRD5A1 and SRD5A2? Also, do the sequences suggest why the two enzymes have radically different pH optima? SRD5A2 has an acidic pH optimum of 5.0-5.5 in in vitro enzyme assays, whereas SRD5A1 has a basic pH optimum of >8.0. In the early days of steroid 5 α -reductase research, the existence of two isozymes was first postulated based on these different pH optima.

We thank the reviewer for the interesting questions. We believe the binding mode of dutasteride is different from that of finasteride at the tail group, which leads to their different selectivity. For finasteride, the tert-butylacetamide tail group forms a hydrogen bond with R114 of SRD5A2 (Fig. 2d). As stated in our revised manuscript, this residue is replaced by a methionine residue (M119) in SRD5A1 (Extended Data Fig. 1), potentially disrupting the hydrogen bond with finasteride and contributing to the selectivity of finasteride for SRD5A2 over SRD5A1. For dutasteride, the trifluoromethyl-benzene tail group, which is much larger than the tert-butyl tail group of finasteride (Fig. 1b), likely occupies a binding site different from that of the tert-butyl tail group of finasteride in SRD5A2. Otherwise, there would be severe clash with residues L20 and L23 (Fig. 1d). As a result, it is possible that dutasteride doesn't form a hydrogen bond with R114 of SRD5A2 as finasteride does. Instead, the trifluoromethyl-benzene group of dutasteride may form strong hydrophobic interactions with relative conserved hydrophobic residues in the ligand entrance area such as L20, L23 and F118 (Extended Data Fig. 1). Therefore, dutasteride doesn't show high enzyme subtype selectivity. However, without a structure of SRD5A2 with dutasteride, we believe such a mechanism is highly speculative and therefore we didn't include it in our manuscript.

Figure R2. Polar residues R114 and E38 involved in ligand-binding of SRD5A2 that are not conserved in SRD5A1.

All residues of SRD5A2 in the catalytic center around the nicotinamide ring of NADP and the C-1 to C-5 ring of DHF are highly conserved in SRD5A1, suggesting that the different optimal pH values of SRD5A1 and SRD5A2 are not due to the different environments of the catalytic centers of these two enzymes. Sequence alignment analysis showed two areas of SRD5A2 around the ligand-binding cavity that are less conserved, the ligand-entrance area where the tert-butylacetamide tail group of DHF resides and the cytosolic loops L1, L3 and L5 that pack

against each other to enclose the binding pocket for NADP (**Extended Data Fig. 5f**). Especially, some polar residues in these two areas are not conserved. For example, in our structural model of SRD5A2 with testosterone, the hydroxyl tail group of testosterone forms a hydrogen bond with R114 (see **Fig. R2** above, left panel), which is replaced by a methionine residue, M119, in SRD5A1 (**Extended Data Fig. 1**). Also, E38 of L1 forms a salt bridge with R105 of L2 to stabilize the conformation of L1 and L2 in the structure of SRD5A2 (see **Fig. R2** above, right panel). E38 of SRD5A2 is replaced by a leucine residue, L43, in SRD5A1 (**Extended Data Fig. 1**). It is possible that there are different polar interaction networks in these areas of SRD5A1 and SRD5A2 that are important for ligand-binding, which may require different optimal environmental pH values. However, it is difficult to propose specific mechanisms without a structure of SRD5A1.

There are many interesting questions regarding ligand recognition and catalysis of 5 α -reductase family. Our study provides the first structural insight into the function of this family and allows us to propose a detailed molecular mechanism of catalysis involving residues E57 and Y91 and highly dynamic cytoplasmic loops. However, one structure is not enough to answer all the questions. More structures of 5 α -reductases including those with physiological substrates are needed for us to gain a comprehensive understanding of this family of enzymes, which may take years of research effort.

3. The work of Makridakis et al. (references 28, 29, and 39 in the present paper) has never been reproduced and thus the contention that gain-of-function mutations exist in SRD5A2 and predispose to prostate cancer is contentious. In the absence of showing enzymatic data that confirm the A49T and A248V alterations are more active, it might be best not to refer to these papers.

We thank the reviewer for pointing it out. We have removed discussion of these gain-of-function mutations in our revised manuscript and included a new **Figure 5**.

Minor Comments

1. The correct symbols for the steroid 5 α -reductase 1 and 2 proteins, respectively are SRD5A1 and SRD5A2, not SRD5 α 1 and SRD5 α 2.

We have changed the protein names to SRD5A1 and SRD5A2 throughout our revised manuscript.

2. Figure 1 legend: "...different R-groups connected in the tails." should read "...different R-groups connected to the amide side chains."

We have revised our manuscript accordingly.

3. Page 6, "...explaining the selectivity of finasteride for human SRD5 α 1 over SRD5 α 2." should read "explaining the selectivity of finasteride for human SRD5A2 over SRD5A1".

We have revised our manuscript accordingly.

Reviewer #4:

The manuscript describes a 2.8 Å crystal structure of a human membrane steroid reductase from the ER responsible for the NADPH dependent reduction of testosterone to dihydrotestosterone. The structure was obtained as a complex with the dualsteric ligand NADP-dihydrofinasteride and provides insight into the manner in which substrates bind the enzyme and the reaction and inhibition mechanisms. Disease causing mutations that raise and lower enzyme activity and ligand selectivity are rationalized based on the structure. The work provides new insights into the mechanism of action and inactivation of a medically important enzyme. Structures of a related bacterial membrane and of a soluble variant of the enzyme are known. While there are similarities, the differences highlighted in the present work represents an important advance for the field. The conclusions and claims of the study are supported by convincing data. The methodology is sound and is of a high standard. However, not enough detail is provided in the methods for the work to be reproduced. The English composition and grammar will need correction.

Points that warrant attention follow.

Discuss the reasons why there are no apo-structures and structures of complexes with one or other substrate? Relate this to the kinetic mechanism of the reaction if known. Is it ping-pong or other, for example?

As we mentioned in our responses to Reviewer #2 and #3, we found in our previous experiments that SRD5A2 without any ligand was not stable during membrane solubilization (see *Fig. R1* above in our responses to Reviewer 2). We believe finasteride or NADP-DHF can stabilize the protein to allow successful protein purification and crystallization. Indeed, with finasteride we could get purified human SRD5A2 in detergent buffers running as a single peak corresponding to a monomeric state on the size exclusion column (**Extended Data Figure 2a**). This is similar to G protein-coupled receptors (GPCRs), which also have large ligand-binding cavities inside the transmembrane domain. Ligand-dependent protein stabilization has been shown to be a fundamental property of GPCRs, and the use of appropriate ligands is critical to GPCR purification and structural characterization ¹⁴.

We use finasteride instead of physiological substrates in our structural study because of its semi-irreversible action. Our MD simulation study suggested a highly flexible nature of cytosolic loops to allow efficient turnover of the reactions with physiological substrates. Therefore, we believe that physiological substrates may not be able to stabilize the enzyme especially the cytosolic loops, resulting in high conformational heterogeneity of the enzyme.

Since SRD5A2 only catalyzes the reduction of single substrate using NADPH as the co-factor, the reaction likely follows the Michaelis-Menten kinetics but not the ping-pong kinetics for multiple-substrate reactions. Previous studies suggested that the cofactor enters the ligand-binding cavity first and leaves after the steroid substrate, which is common among NADPH-dependent enzymes ¹⁰.

25 The binding cavity is not ‘inside the membrane’

We have changed it to 'inside the transmembrane domain'.

28 Do the simulations tell us anymore that what is already available from crystallographic B-factors and missing residues?

Crystallographic B-factors and missing residues in the crystal structure are heavily affected by crystal packing interactions and crystallization conditions, which do not reflect physiological conditions. They can only suggest a possibility of structural flexibility. For example, the cytosolic loops L1, L3 and L5 of SRD5A2 mediate crystal packing interactions among symmetric molecules (**Extended Data Fig. 2**). The missing residues in L1 is likely due to the steric hindrance caused by crystal packing. Also, the cytosolic loop L5 of one SRD5A2 molecule interacts with that of another symmetric molecule in the crystal structure. Therefore, its B-factors are lower than the B-factors of the C-loop (**Extended Data Figure 8**). However, we observed larger conformational changes of L5 compared to the C-loop in the simulations. More importantly, MD simulations were performed using single SRD5A2 molecule reconstituted in physiologically relevant lipid bilayers and we followed protein conformational changes over the time course of simulation. Crystallographic B-factors and missing residues cannot provide detailed information regarding protein conformational changes as exemplified in **Extended Data Figure 6a**. Therefore, we analyzed protein dynamics mainly based on our MD simulation results. We only used crystallographic B-factors as supporting evidence.

38 List the other lipid substrates.

As an important breakthrough in the field, SRD5A3 has been identified to reduce polyprenol to dolichol in human¹⁷. We have specified polyprenols as the other lipid substrates in our revised manuscript.

41 Explain how these enzymes are involved in N-linked glycosylation.

SRD5A3 has been identified to reduce polyprenol to dolichol in human¹⁷. Dolichol is required for the synthesis of dolichol-linked monosaccharides and oligosaccharide precursors to be used in N-linked glycosylation. We have cited this study in our manuscript.

46 inefficient/insufficient?

We have changed it to 'insufficient' in our revised manuscript.

56 The link to Covid-19 is not clear and should be explained.

We now state that 'androgen receptor signaling can lead to the expression of transmembrane serine protease 2 (TMPRSS2), which is required for the entry of SARS-CoV-2 and other coronaviruses in the host cells. Therefore, androgen signaling has recently been linked to COVID-19 disease severity, explaining why males are more prone to severe COVID-19 symptom' in the Introduction with appropriate citations.

79 The missing residues in L1 need to be referred to in the simulations work.

We have mentioned in the first line of the 'Molecular modelling and MD simulations' section in Methods that 'SRD5α2 models were built based on the solved crystal structure to incorporate unresolved residues 1-4 (N-terminus) and 39-43 (loop L1) using Modeller'.

80 Explain dualsteric for the benefit of the general reader.

The concept of 'dualsteric ligands' is mainly used for G protein-coupled receptors, which refers to ligands that can occupy two separate binding sites simultaneously. We borrowed this concept to indicate the two different moieties in NADP-DHF adduct that occupy distinct sites. We now state that 'A dualsteric ligand that occupies two different binding sites was modeled as an adduct of finasteride and NADPH in the structure".

98 Shorter than 2 Å is not useful. State actual bond length.

We have specified the distance to be $\sim 1.5 \text{ \AA}$ in the revised manuscript.

Fig. 2 Show the entire molecule in b to match that in a.

Show membrane boundaries in c.

State criteria for a hydrogen bond.

We have revised **Figure 2** and the legend as suggested by the reviewer.

110 Provide values for binding constants and off rates.

We have provided those values in the second paragraph of section 'Intermediate adduct formed between finasteride and NADPH' in our revised manuscript.

189 Use the same color scheme for the two NADPs and finasterides in f.

We have changed the color scheme as suggested by the reviewer.

198 Explain why finasteride does not inhibit 100% in e.

Finasteride binding to SRD5A2 is a slow process¹⁰. In our experiments, finasteride and testosterone were added at the same time. It is highly possible that there were SRD5A2 molecules not occupied by finasteride during the experiments. Similar, in a previous study, the authors also reported that finasteride could not instantly inhibit all SRD5A2 activity¹⁸.

218 Has the pKa of E57 been estimated in support of the assumed protonation at pH~5?

The side chain of E57 is protonated in the crystal structure since it forms hydrogen bonds with the C-3 carbonyl group and the N-4 amine group of DHF. In our structural model of SRD5A2 with testosterone, the side chain of E57 also forms a hydrogen bond with the C-3 carbonyl group, suggesting a protonated state. We did not assume that E57 was protonated based on its estimated pKa. We suggest that E57 protonation is important for the catalysis, and an acidic pH favors the protonation of E57 and thus favors the reaction. This is based on our mutagenesis data (**Fig. 3e**) and structural comparison with the steroid 5 β -reductase AKR1D1, in which a protonated glutamic acid residue, Glu120, forms a hydrogen bond with the C-3 carbonyl atoms of its steroid substrates to facilitate hydride transfer¹⁹.

219 could/does

We have changed the sentence in the third paragraph of section 'Potential mechanisms of SRD5A2 catalysis and inhibition' as 'substitution of E57^{TM2} to a less acidic glutamine residue compromised enzyme activity'.

231 It is important to alert the reader to the fact that the crystal structure is missing 5 residues in L1 and that these have been modelled back in for simulations work.

We have mentioned this in Methods. In addition, we now state in the second paragraph of section 'Structural dynamics of SRD5A2 for catalysis' that 'Unresolved residues 1-4 at the N-terminus and 39-43 in L1 were incorporated into the SRD5A2 models in simulations.'

Fig. 6 The superpose is not obvious in a. Show membrane boundaries in a and b.

We have revised **Figure 6** to show superimposition of the aligned transmembrane helices (**Fig. 6a and c**) and show the membrane boundaries in panels a and b.

Table 1. Three decimal places are not justified for cell dimensions.

We used HKL2000 to process the diffraction data and we simply reported the cell dimension values in the final file containing the structure factor data (.sca file) generated by HKL2000. Nevertheless, we have changed the values with one decimal place in the revised manuscript.

Methods section must be rewritten with enough detail for the work to be reproduced. Concentrations, volumes, order of addition, mixing, time, temperatures, numbers of dounce homogenizations, etc must be reported. Reasons for including various additives (IAA, benzamidine, calcium, etc) should be explained.

We have revised our Methods section to include detailed information of sample preparation and crystallization as suggested by the reviewer.

478 How was the protein concentrated? What is the composition of the final buffer?

As stated in our revised Methods section, the protein was concentrated using protein concentrators with a molecular-weight cutoff (MWCO) of 50K Dalton by centrifugation, and the final protein buffer contained 20 mM HEPES pH7.5, 100 mM NaCl, 1.0 μ M finasteride, 0.01% LMNG and 0.001% CHS.

480 LCP reference is incorrect. It should be the Nature Protocols 2009 reference.

We cited the paper from Dr. Martin Caffrey published on Annual Review of Biophysics at 2009 because this paper discussed principals and general applications of the LCP method. The paper from Dr. Caffrey on Nature Protocols provided a detailed protocol for this method. We have cited both papers in our revised manuscript.

483 What spacer thickness was used?

As stated in our revised Methods section, we used 3M tape 9500CC with holes as spacers. The adhesive thickness of 3M tape 9500CC is about 0.058 mm.

527 Was E57 in its standard state. Indicate what the standard state is and how this relates to the proposed mechanism.

Yes. E57 was modelled in its standard state (anionic form) in the simulations. In the crystal structure, E57 is protonated, possibly as a consequence of the low pH of crystallization conditions (pH=5) and the presence of NADP-DHF. The protonated E57 forms hydrogen bonds with DHF, which plays a key role in the proposed catalytic mechanism as shown in **Figure 3**. However, in our simulations, we didn't include either finasteride or testosterone. We mainly focused on the structural dynamics of SRD5A2 with and without NADP in the simulations. The protonation states of all ionizable residues were calculated at pH=7 by solving Poisson-Boltzmann equation in continuum electrostatics models ($\epsilon_{\text{protein}}=20$; $\epsilon_{\text{solvent}}=80$) using APBS (see Methods). Based on these results, E57 was modelled in its standard state (anionic form) in both *apo* and *nap* states in the simulations. Our simulation study was not intended to investigate the hydride transfer mechanism of the reaction. But we agree this is an interesting point for further study. It is possible that the binding of NADPH and testosterone may change the environment of the catalytic centre and thus the protonation state of E57.

561 mg of what? How was it quantified?

For each reaction, we used membrane fraction containing 0.3 mg total protein. We used the Bradford assay to quantify total protein in the membrane fraction. We have revised our Methods section to include this information.

563 Explain how the negative control was run.

For the negative control, 0.5 mM finasteride was added into the membrane mix with NADPH and testosterone.

566 Peak area of what? What signal was quantified? Is the response the same for the two analytes?

The signals to quantify came from the intensity of ionized dihydrotestosterone and testosterone in the liquid chromatography–mass spectrometry (LC-MS) assay. We integrated areas of peaks corresponding to dihydrotestosterone and testosterone in the LC-MS chromatograms to estimate the amount of those two analytes in the reaction. The ratios of dihydrotestosterone and testosterone were calculated as indicators of relative enzyme activity. We have included this information in the revised manuscript and provided the raw data in the source data file we submitted together with our revised manuscript. This method has been widely used in the studies of steroid hormones²⁰.

ED Fig 4 d and e switched.

We have corrected this mistake.

State program used for 2D view.

We now state in the figure legend that the 2D view was prepared by LigPlot⁺.

Is there any significance attached to the difference seen between the WT and the two mutants in expression levels in h?

No. The **Extended Data Figure 5h** was just to show that mutants E57Q and Y91F were well expressed in the cells so the decreased enzyme activity was not due to the loss of protein expression.

References

- 1 Sjodt, M. *et al.* Structure of the peptidoglycan polymerase RodA resolved by evolutionary coupling analysis. *Nature* **556**, 118-121, doi:10.1038/nature25985 (2018).
- 2 Haas, J. *et al.* Continuous Automated Model EvaluatiOn (CAMEO) complementing the critical assessment of structure prediction in CASP12. *Proteins* **86 Suppl 1**, 387-398, doi:10.1002/prot.25431 (2018).
- 3 Kim, D. E., Chivian, D. & Baker, D. Protein structure prediction and analysis using the Robetta server. *Nucleic Acids Res* **32**, W526-531, doi:10.1093/nar/gkh468 (2004).
- 4 Soding, J., Biegert, A. & Lupas, A. N. The HHpred interactive server for protein homology detection and structure prediction. *Nucleic Acids Res* **33**, W244-248, doi:10.1093/nar/gki408 (2005).
- 5 Kallberg, M. *et al.* Template-based protein structure modeling using the RaptorX web server. *Nat Protoc* **7**, 1511-1522, doi:10.1038/nprot.2012.085 (2012).
- 6 Waterhouse, A. *et al.* SWISS-MODEL: homology modelling of protein structures and complexes. *Nucleic Acids Res* **46**, W296-W303, doi:10.1093/nar/gky427 (2018).
- 7 Mariani, V., Biasini, M., Barbato, A. & Schwede, T. IDDT: a local superposition-free score for comparing protein structures and models using distance difference tests. *Bioinformatics* **29**, 2722-2728, doi:10.1093/bioinformatics/btt473 (2013).
- 8 Di Costanzo, L., Drury, J. E., Christianson, D. W. & Penning, T. M. Structure and catalytic mechanism of human steroid 5beta-reductase (AKR1D1). *Mol Cell Endocrinol* **301**, 191-198, doi:10.1016/j.mce.2008.09.013 (2009).
- 9 Chen, M. & Penning, T. M. 5beta-Reduced steroids and human Delta(4)-3-ketosteroid 5beta-reductase (AKR1D1). *Steroids* **83**, 17-26, doi:10.1016/j.steroids.2014.01.013 (2014).
- 10 Bull, H. G. *et al.* Mechanism-Based Inhibition of Human Steroid 5 α -Reductase by Finasteride: Enzyme-Catalyzed Formation of NADP–Dihydrofinasteride, a Potent Bisubstrate Analog Inhibitor. *Journal of the American Chemical Society* **118**, 2359-2365, doi:10.1021/ja953069t (1996).
- 11 Schmidt, L. J. & Tindall, D. J. Steroid 5 alpha-reductase inhibitors targeting BPH and prostate cancer. *J Steroid Biochem Mol Biol* **125**, 32-38, doi:10.1016/j.jsbmb.2010.09.003 (2011).
- 12 Tian, G., Mook, R. A., Jr., Moss, M. L. & Frye, S. V. Mechanism of time-dependent inhibition of 5 alpha-reductases by delta 1-4-azasteroids: toward perfection of rates of time-dependent inhibition by using ligand-binding energies. *Biochemistry* **34**, 13453-13459, doi:10.1021/bi00041a024 (1995).
- 13 Aggarwal, S., Thareja, S., Verma, A., Bhardwaj, T. R. & Kumar, M. An overview on 5alpha-reductase inhibitors. *Steroids* **75**, 109-153, doi:10.1016/j.steroids.2009.10.005 (2010).
- 14 Zhang, X., Stevens, R. C. & Xu, F. The importance of ligands for G protein-coupled receptor stability. *Trends Biochem Sci* **40**, 79-87, doi:10.1016/j.tibs.2014.12.005 (2015).
- 15 Caffrey, M. Crystallizing membrane proteins for structure determination: use of lipidic mesophases. *Annu Rev Biophys* **38**, 29-51, doi:10.1146/annurev.biophys.050708.133655 (2009).
- 16 Peng, H. M. *et al.* Expression in Escherichia Coli, Purification, and Functional Reconstitution of Human Steroid 5alpha-Reductases. *Endocrinology* **161**, doi:10.1210/endocr/bqaa117 (2020).

- 17 Cantagrel, V. *et al.* SRD5A3 is required for converting polyprenol to dolichol and is mutated in a congenital glycosylation disorder. *Cell* **142**, 203-217, doi:10.1016/j.cell.2010.06.001 (2010).
- 18 Faller, B., Farley, D. & Nick, H. Finasteride: a slow-binding 5 alpha-reductase inhibitor. *Biochemistry* **32**, 5705-5710, doi:10.1021/bi00072a028 (1993).
- 19 Di Costanzo, L., Drury, J. E., Penning, T. M. & Christianson, D. W. Crystal structure of human liver Delta4-3-ketosteroid 5beta-reductase (AKR1D1) and implications for substrate binding and catalysis. *J Biol Chem* **283**, 16830-16839, doi:10.1074/jbc.M801778200 (2008).
- 20 Srivilai, J. *et al.* A new label-free screen for steroid 5alpha-reductase inhibitors using LC-MS. *Steroids* **116**, 67-75, doi:10.1016/j.steroids.2016.10.007 (2016).

REVIEWERS' COMMENTS

Reviewer #1 (Remarks to the Author):

The authors have addressed all the comments and this referee is satisfied with their revision.

Reviewer #2 (Remarks to the Author):

The authors have fully addressed my questions. The manuscript is recommended for publication.

Reviewer #3 (Remarks to the Author):

The authors have satisfactorily addressed the issues raised in the initial critique.

Reviewer #4 (Remarks to the Author):

The revised manuscript and the responses address the points raised by Reviewer #4. One minor point. The thickness information requested refers to the tape itself and, by extension, to well depth. The response refers to an adhesive thickness not the tape thickness. Tape thickness should be verified and corrected as necessary.

Manuscript ID: NCOMMS-20-26845-T

Title: **Structure of human steroid 5 α -reductase 2 with anti-androgen drug finasteride**

REVIEWERS' COMMENTS

Reviewer #1 (Remarks to the Author):

The authors have addressed all the comments and this referee is satisfied with their revision.

Reviewer #2 (Remarks to the Author):

The authors have fully addressed my questions. The manuscript is recommended for publication.

Reviewer #3 (Remarks to the Author):

The authors have satisfactorily addressed the issues raised in the initial critique.

Reviewer #4 (Remarks to the Author):

The revised manuscript and the responses address the points raised by Reviewer #4.
One minor point. The thickness information requested refers to the tape itself and, by extension, to well depth. The response refers to an adhesive thickness not the tape thickness. Tape thickness should be verified and corrected as necessary.

The adhesive refers to the tape. The adhesive thickness represents the well depth (~0.14mm). We have specified it in the Methods section.